# Long-Context Modeling with Dynamic Hierarchical Sparse Attention for Memory-Constrained LLM Inference

Siheng Xiong [1] [*]   Joe Zou [2]   Faramarz Fekri [1]   Yae Jee Cho [2]

## Abstract

The quadratic cost of attention limits the scalability of long-context LLMs, especially under limited hardware memory budgets. While attention is often sparse, existing static sparse methods cannot adapt to task- or input-dependent variations, and recent dynamic approaches rely on predefined templates or heuristics that may sacrifice generality. We propose Dynamic Hierarchical Sparse Attention (DHSA), a data-driven framework that predicts attention sparsity online while keeping the LLM backbone frozen. DHSA performs hierarchical routing by estimating importance at the chunk level and propagating it to token-level interactions, preserving causally important dependencies while enabling efficient sparsification. Across Needle-in-a-Haystack test, LongBench and RULER, DHSA maintains near-dense accuracy in highly sparse regimes, achieving 12–20% relative accuracy gains over Block Sparse Attention at comparable prefill cost. With a memory-efficient tiled backend, DHSA delivers up to $10\times$ prefill speedup at 128K context length. On LLaMA-3.1-8B (4-bit), DHSA scales to 100K context on a single 24GB GPU, where dense attention fails. We provide complementary GPU and CPU backends, enabling DHSA to run across diverse hardware environments and multiple open-weight model families. These results demonstrate DHSA as an efficient and adaptable solution for memory-constrained long-context LLM inference. Code is available at: https://github.com/xiongsiheng/DHSA.

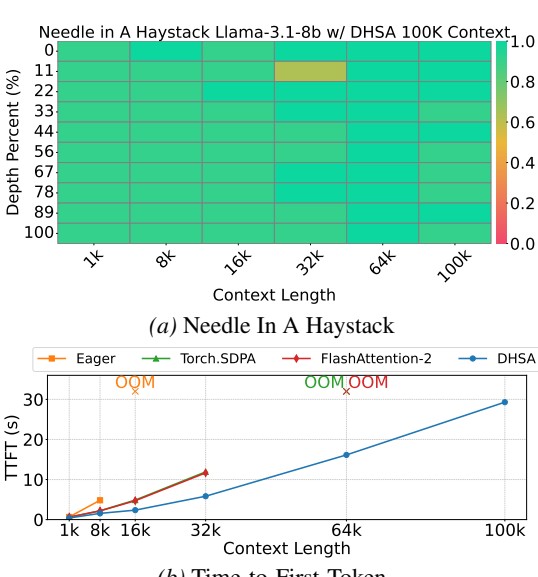

*(a)* Needle In A Haystack

*(b)* Time-to-First-Token

*Figure 1.* Performance of DHSA on LLaMA-3.1-8B (4-bit). (a) Needle-in-a-Haystack accuracy under a fixed density of 6.25%, where DHSA preserves high retrieval accuracy up to 100K context length. (b) Time-to-first-token (TTFT) on a single NVIDIA RTX 3090 (24 GB). DHSA scales to long contexts while remaining feasible in memory-constrained settings.

## 1. Introduction

Long-context modeling expands the capabilities of large language models (LLMs) to a wide range of real-world applications, including proofreading and summarizing long documents, analyzing personal histories, and maintaining extended context in personal assistants. However, the quadratic complexity of the attention mechanism (Vaswani et al., 2017) leads to prohibitive computational costs, limiting the scalability of long-context tasks, particularly under limited hardware memory budgets.

Prior research has shown that attention matrices in LLMs are highly sparse (Child, 2019), motivating the development of static sparse attention methods such as Longformer (Beltagy et al., 2020) and BigBird (Zaheer et al., 2020) for more efficient inference. However, later studies demonstrated that attention distributions vary significantly across different tasks and inputs (Jiang et al., 2024). Such dynamic

[1]Georgia Institute of Technology. [2]Google. [*]Part of the work was done while Siheng Xiong was an intern at Google. Correspondence to: Siheng Xiong <sxiong45@gatech.edu>, Yae Jee Cho <yaejeecho@google.com>.

*Proceedings of the 43rd International Conference on Machine Learning*, Seoul, South Korea. PMLR 306, 2026. Copyright 2026 by the author(s).

nature of the attention patterns limits the effectiveness of static sparse methods in long-context settings, as they often incur noticeable performance drops (Figure 13). Consequently, if the sparse attention patterns could be predicted in an input-adaptive dynamic manner, long-context LLMs can substantially reduce resource computational costs while maintaining accuracy. Recent work has explored dynamic sparsity from different angles. MInference (Jiang et al., 2024) accelerates prefill via calibrated sparsity templates, while H2O (Zhang et al., 2023) targets decoding through heuristic KV-cache eviction. Despite their effectiveness, both rely on manually designed patterns or rules, which limits their ability to capture highly input-dependent attention sparsity.

To address these challenges, we propose **Dynamic Hierarchical Sparse Attention (DHSA)**, a *drop-in sparse attention module* for standard decoder-only Transformers. DHSA *keeps the LLM backbone frozen*: given per-layer queries and keys, it performs routing to produce compact token index sets, which the sparse attention backend consumes to prune computation during prefill. Unlike template- or heuristic-based methods, DHSA learns content-adaptive sparsity online from token embeddings, delivering speedups across tasks without manual tuning.

Our contributions are summarized as follows:

- **Hierarchical routing formulation.** We recast sparsification as *chunk-level routing*: predicting a chunk–chunk similarity matrix and using it to route attention to a compact set of relevant tokens, preserving the most impactful query–key interactions while maintaining causal semantics (§3.1).

- **Content-aware segmentation with length-robust representations.** A lightweight boundary predictor uses token keys to *dynamically chunk* sequences. The predictor is shared across layers and adds only a linear-time pass in sequence length. Each chunk is then encoded with a length-normalized pooling scheme to produce stable chunk queries/keys for reliable similarity estimation (§3.2, §3.3).

- **Practical model- and hardware-agnostic sparse attention module.** DHSA is implemented as a drop-in module that interposes between query/key projections and the attention backend, performing token-level selection via routed index sets. It supports standard decoder-only Transformers on both GPUs and CPUs while keeping the LLM backbone frozen (§3.4).

Extensive experiments on Needle-in-a-Haystack test, Long-Bench (Bai et al., 2024) and RULER (Hsieh et al., 2024) show that DHSA preserves near-dense accuracy in highly sparse regimes, yielding 12–20% relative accuracy gains

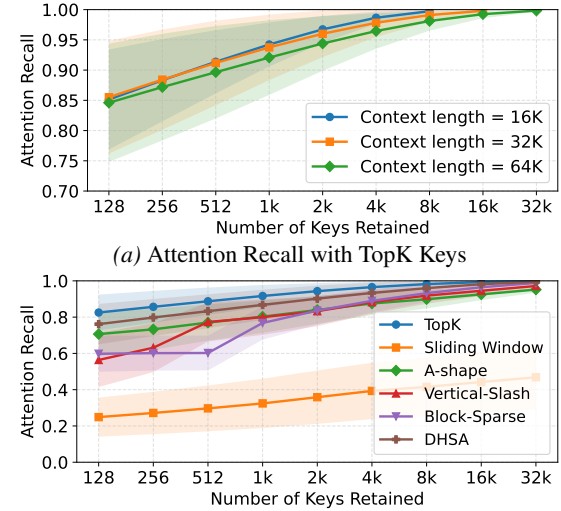

*(a) Attention Recall with TopK Keys*

*(b) Attention Recall across Sparsity Patterns*

*Figure 2.* We analyze attention sparsity on LLaMA-3.1-8B (4-bit). (a) More than 95% of the attention mass can be recovered by retaining only a small subset of keys. (b) DHSA consistently achieves higher recall than the other sparse patterns. See Appendix B, Figure 11 and Figure 12 for theoretical analysis and additional empirical comparisons.

over Block Sparse Attention (Guo et al., 2024) at comparable prefill cost (Figure 5). DHSA is compatible with multiple open-weight model families and both GPU and CPU backends (Table 2), making it an efficient and adaptable solution for memory-constrained long-context inference.

## 2. Predicting Attention Sparsity in Long Contexts

**Long-Context Attention is Sparse.** The quadratic cost of attention makes scaling LLMs to long contexts prohibitively expensive, especially in resource-constrained settings. However, a closer examination of attention distributions shows that much of this computation is unnecessary (Child, 2019). For example, as shown in Figure 2a, preserving only a small fraction of keys retains more than 95% of the total attention mass in LLaMA-3.1-8B (4-bit). In practice, each query token attends to only a limited subset of the sequence. These findings indicate that efficient long-context inference is achievable by accurately identifying and retaining the most important token interactions.

**Attention Sparsity Requires Input-Adaptive Prediction.** Although long-context attention is sparse, the locations of salient tokens vary significantly across inputs, as relevance depends on the current query. Existing approaches often rely on static templates (e.g., sliding window (Beltagy et al., 2020), A-shape (Han et al., 2024)) or predefined heuristics (e.g., Vertical-Slash (Jiang et al., 2024)), which limit performance on various tasks. At the same time, attention maps re-

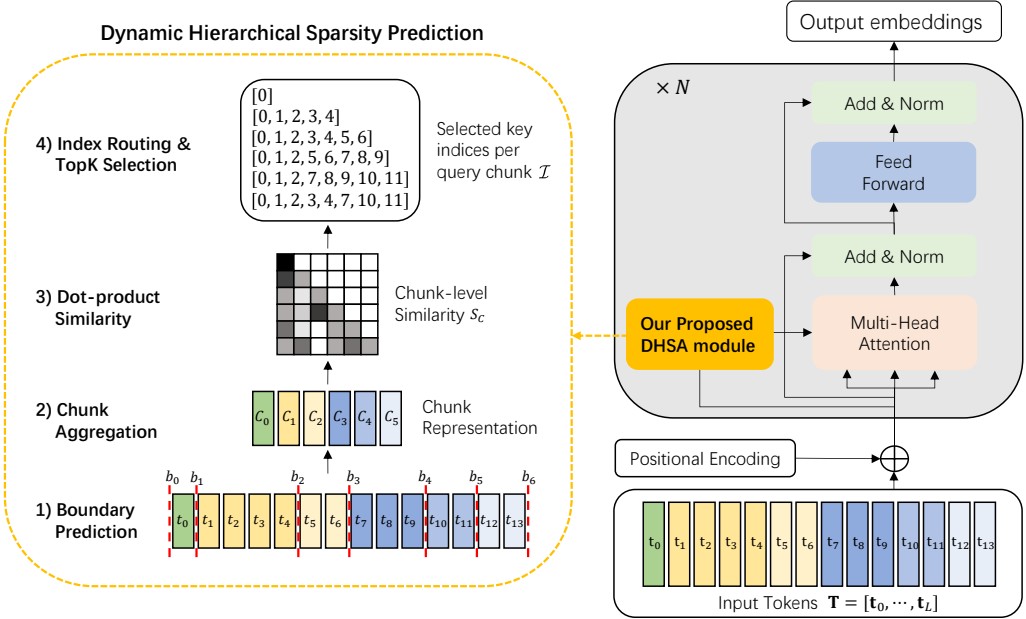

*Figure 3.* Overview of our proposed Dynamic Hierarchical Sparse Attention (DHSA) framework.

veal structural regularities: salient tokens cluster into spans such as sentences, paragraphs, or code blocks. However, existing dynamic methods (e.g., Block-Sparse (Guo et al., 2024)) disregard this structure by partitioning sequences into fixed-length chunks.

To this end, we propose **Dynamic Hierarchical Sparse Attention (DHSA)** that leverages the hierarchy that occurs from the structural regularities of attention maps for more efficient and performative long context modeling. DHSA predicts the importance of input tokens online by first performing dynamic chunking (example shown in Figure 11b) based on our proposed dynamic boundary detection method (Section 3.2) and then using chunk-level similarity to guide token-level sparsity (Section 3.3). As shown in Figure 11a, DHSA produces more flexible and adaptive patterns than existing methods. Furthermore, our comparisons in the attention recall shown in Figure 2b demonstrate that DHSA consistently achieves higher recall and closely approaches TopK performance.

## 3. Dynamic Hierarchical Sparse Attention

Our proposed DHSA is a drop-in module integrated into each of the $N$ Transformer layers of an LLM (see Figure 3). It takes the token embeddings at the current layer as the input and outputs a sparse attention pattern. To enable efficient prediction, we first group consecutive tokens into *chunks*. The core idea is to leverage **chunk-level similarity** to inform **token-level sparsity** prediction. This requires addressing two key challenges: (1) fixed-size chunking is

too rigid to capture **content shifts** across varying inputs, and (2) average pooling poorly handles **variable-length chunks**. We resolve these with the solutions detailed in Sections 3.2 and 3.3, and validate that DHSA addresses both challenges in our ablation study (Table 4). First, in Section 3.1 we give an overview of the steps for hierarchical sparsity prediction.

### 3.1. Hierarchical Sparsity Prediction

Given a token sequence $\mathbf{T} = [\mathbf{t}_0, \mathbf{t}_1, \ldots, \mathbf{t}_{L-1}]$ of length $L$, we define a token-level sparsity mask $\mathbf{M} \in \{0, 1\}^{L \times L}$, where $\mathbf{M}_{i,j} = 1$ indicates that the interaction between tokens $\mathbf{t}_i$ and $\mathbf{t}_j$ is preserved, and $\mathbf{M}_{i,j} = 0$ indicates that it is skipped. Predicting the full matrix $\mathbf{M}$ directly would require scoring all $L \times L$ token pairs, which is computationally prohibitive for long contexts. Instead, we adopt a two-step hierarchical approach:

**Step 1** **Chunk-level prediction**: We partition the entire token sequence $\mathbf{T}$ into $N_c$ non-overlapping chunks $\{\mathbf{C}_0, \mathbf{C}_1, ..., \mathbf{C}_{N_c-1}\}$, defined by the boundary indices $\mathcal{B} = \{b_0, b_1, ..., b_{N_c}\}$, where $0 = b_0 < b_1 < ... < b_{N_c} = L$. Each chunk $\mathbf{C}_k$, $k \in [0, ..., N_c - 1]$ contains the consecutive tokens indexed from $b_k$ to $b_{k+1}$. We then construct a chunk-level similarity matrix $\mathbf{S}_c \in \mathbb{R}^{N_c \times N_c}$, where the $l^{th}$ row and $k^{th}$ column element of $\mathbf{S}_c$, i.e., $(\mathbf{S}_c)_{l,k}$ represents the predicted importance of interactions between chunks $\mathbf{C}_l$ and $\mathbf{C}_k$.

**Step 2** **Token-level selection**: We use $\mathbf{S}_c$ to *route* each query chunk to a compact set of key token indices.

Concretely, for each query chunk, we rank key chunks according to chunk-level similarity scores and progressively expand them into their constituent token indices, while enforcing causal constraints. This yields, for each query chunk $k$, an order set of selected key indices $\mathcal{I}_k \subseteq \{0, \ldots, L-1\}$. The expansion is terminated once $|\mathcal{I}_k|$ exceeds the per-query token budget $N_b$.

By operating on chunk-level scores and directly producing compact index sets, DHSA avoids constructing dense token-level similarity matrices. This hierarchical routing mechanism substantially reduces both computation and memory overhead compared to naive token-pair scoring, as demonstrated in Figure 1b. Next, we describe how the boundary indices $\mathcal{B}$ are determined in Section 3.2.

### 3.2. Dynamic Boundary Detection

We propose a dynamic chunking strategy that adaptively determines boundary indices $\mathcal{B}$ based on the input sequence $\mathbf{T}$. We formulate chunking as a **boundary detection** problem, where the goal is to decide whether each token position marks the *end of a chunk*. The boundary predictor is trained using automatically derived soft labels from dense attention patterns, as described below. At inference time, DHSA predicts boundary scores from local key-vector windows and finalizes chunk boundaries through thresholding and non-maximum suppression (NMS). Formally, for each position $i \in [0, L-1]$, we define a boundary indicator function $\delta(i) = 1$ if $i = b_k$ for some $k$; otherwise, $\delta(i) = 0$. We estimate this indicator using a neural network with three components also shown in Figure 4:

**Encoder.** For each candidate position $i$, we extract two local windows:

$$\begin{aligned}
\mathbf{k}_{\text{left}} &= \text{Pool}(\text{Enc}([\mathbf{k}_{i-w+1}, \cdots, \mathbf{k}_i])), \\
\mathbf{k}_{\text{right}} &= \text{Pool}(\text{Enc}([\mathbf{k}_{i+1}, \cdots, \mathbf{k}_{i+w}])),
\end{aligned} \tag{1}$$

where $w$ is the window size and $\mathbf{k}_i$ denotes the key vector of token $i$. The encoder is a standalone self-attention module that processes the input key vectors through learned projections to queries, keys, and values, followed by pooling to obtain a fixed-length representation. Its parameters are independent of the base LLM. The detailed implementation of the encoder is provided in Appendix A.2.

**Feature Fusion.** Given $\mathbf{k}_{\text{left}}$ and $\mathbf{k}_{\text{right}}$, we construct the feature vector:

$$\begin{aligned}
\mathbf{h}_i = [&\mathbf{k}_{\text{left}}, \ \mathbf{k}_{\text{right}}, \ |\mathbf{k}_{\text{left}} - \mathbf{k}_{\text{right}}|, \\
&\mathbf{k}_{\text{left}} \odot \mathbf{k}_{\text{right}}, \ \text{sim}(\mathbf{k}_{\text{left}}, \mathbf{k}_{\text{right}})]
\end{aligned} \tag{2}$$

where $\odot$ denotes element-wise multiplication and $\text{sim}(\cdot, \cdot)$ denotes cosine similarity between vectors. Further ratio-

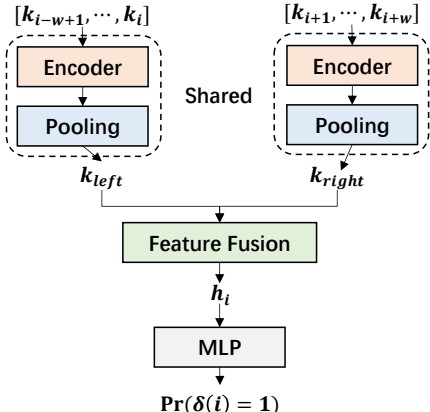

*Figure 4.* Architecture of boundary predictor, consisting of a shared encoder, a feature fusion layer, and an MLP.

nale and analyses of this fusion choice are provided in Appendix A.2.

**MLP.** The fused feature $\mathbf{h}_i$ is passed through a two-layer fully connected network, denoted as MLP, yielding the probability that position $i$ is a boundary:

$$\Pr(\delta(i) = 1) = \text{MLP}(\mathbf{h}_i). \tag{3}$$

**Automatic Labeling.** To train the boundary predictor without manual annotations, we derive soft labels from dense attention patterns of the base model. Specifically, we compare the accumulated attention mass over the left and right local windows around each candidate position; a sharp difference indicates a likely boundary. This provides dense local supervision while avoiding end-to-end optimization through discrete boundary decisions. Full details are provided in Appendix A.3.

**Inference Pipeline.** Given the sequence $\mathbf{T} = [\mathbf{t}_0, \ldots, \mathbf{t}_{L-1}]$, we extract their key vectors $\mathbf{K} = \{\mathbf{k}_i\}_{i=0}^{L-1}$. For each position $i$, the predictor takes as inputs two local context windows $\mathbf{K}_{i-w+1:i}$ and $\mathbf{K}_{i+1:i+w}$, and outputs a probability $\Pr(\delta(i) = 1)$. After obtaining boundary scores for all positions, we filter low-confidence candidates with a minimal threshold and apply NMS, retaining only the highest-scoring boundary within each local window to avoid redundant or overly dense boundaries. This produces the boundary set $\mathcal{B} = \{b_0, b_1, \ldots, b_{N_c}\}$, where each $b_k$ marks the end of a chunk. Unless otherwise specified, we use the same thresholding and NMS settings across model families. Additional training and inference details for our boundary predictor are provided in Appendices A.3 to A.5.

### 3.3. Robust Chunk Representation

Now that we have the boundaries $\mathcal{B}$, we aggregate the token embeddings to form chunk representations. We identify

two main challenges arise in this process: (1) average pooling after padding is problematic, as zero embeddings from padding dilute the average, and (2) average pooling is sensitive to chunk length. To address these issues, we compute the prefix sum of embeddings and divide by the actual chunk length, followed by **length normalization**:

$$\mathbf{q}_{\mathbf{C}_k} = \frac{\sqrt{b_{k+1} - b_k}}{b_{k+1} - b_k} \sum_{i \in [b_k, b_{k+1})} \mathbf{q}_i,$$

$$\mathbf{k}_{\mathbf{C}_k} = \frac{\sqrt{b_{k+1} - b_k}}{b_{k+1} - b_k} \sum_{i \in [b_k, b_{k+1})} \mathbf{k}_i. \tag{4}$$

where chunk $\mathbf{C}_k$ spans the token interval $[b_k, b_{k+1})$, and $\mathbf{q}_i, \mathbf{k}_i$ are the token-level query and key vectors, respectively. Stacking the chunk representations yields $\mathbf{Q}_c = [\mathbf{q}_{\mathbf{C}_1}, \ldots, \mathbf{q}_{\mathbf{C}_{N_c}}]^\top$ and $\mathbf{K}_c = [\mathbf{k}_{\mathbf{C}_1}, \ldots, \mathbf{k}_{\mathbf{C}_{N_c}}]^\top$. The chunk-level similarity matrix is then given by $\mathbf{S}_c = \mathbf{Q}_c \mathbf{K}_c^\top$.

### 3.4. Accelerating LLM Inference through DHSA

DHSA is designed to accelerate long-context LLM inference by reducing the quadratic attention cost during the *prefill* stage. We focus on long-context, *prefill-dominated* workloads. Optimizing the decoding stage for long outputs involves additional orthogonal considerations, such as KV-cache management and token sampling, and is therefore left to future work. In principle, the same sparse attention mechanism can also be applied during decoding by restricting the keys attended by each newly generated token.

**Workflow.** Given a prompt of length $L$, DHSA operates in a lightweight routing–and–compute pipeline. First, a boundary predictor partitions the input sequence into variable-length chunks and produces a chunk-level similarity matrix. Based on these scores, DHSA routes attention for each query chunk to $N_b$ relevant key tokens. These routed index sets are then consumed by a sparse attention backend, which computes exact causal attention only over the selected keys.

**Implementation.** To enable efficient execution on modern hardware, we *decouple* the partitioning of queries and keys (Algorithm 1). Queries are processed in fixed-size row blocks, while keys are selected via DHSA's routing mechanism. This design preserves a regular computation structure while allowing fine-grained, token-level sparsity on the key side. Based on this abstraction, we provide two complementary attention backends. A PyTorch SDPA backend (Algorithm 2) offers broad compatibility across model families and platforms, including CPU. A tiled online-softmax backend (Algorithm 3) adopts a streaming softmax, processing selected keys in tiles without materializing attention matrices, and achieves higher efficiency on supported GPUs. Both backends consume the same routed index sets and

share the same workflow. We also introduce additional memory optimizations for very long input sequences (e.g., $\geq 64K$ tokens). Details are provided in Appendix C.

**Complexity.** By restricting attention computation to the selected tokens, DHSA reduces the per-layer attention cost from $O(L^2)$ to approximately $O(L \cdot N_b)$. We define the *token density* as $N_b/L$. A detailed theoretical analysis is provided in Appendix B.6.

## 4. Experiments

In this section, we evaluate DHSA in terms of both effectiveness and efficiency.

**Implementation Details.** To study long-context inference under memory constraints, we select Llama-3.1-8B-Instruct (Dubey et al., 2024), Qwen2.5-3B-Instruct (Yang et al., 2024a), and gemma-2-2b-it (Team et al., 2024). We apply *4-bit quantization* for Llama-3.1-8B-Instruct, and *torch.bfloat16* precision for Qwen2.5-3B-Instruct and Gemma-2-2B-it. Additional results on Llama-3.1-8B-Instruct (BF16) and Qwen2.5-14B-Instruct (4-bit) are provided in Tables 8 and 11, showing that DHSA applies beyond the default setting and maintains both accuracy and latency advantages on larger or higher-precision backbones. All GPU-related experiments are conducted on a single NVIDIA RTX 3090 GPU. For CPU experiments, we use an Intel Core 5 120U processor. To ensure stable results, all experiments use greedy decoding.

Our method is implemented in PyTorch with Hugging Face Transformers, with two complementary attention backends. For training the boundary detector, we use Long Data Collections, TriviaQA (Joshi et al., 2017), and ChatQA2 (Xu et al., 2025). The boundary predictor is trained separately for each backbone family, since it operates on backbone-specific key representations. This training only updates the lightweight boundary predictor; the LLM backbone remains frozen throughout. Although these datasets are primarily QA-oriented, the boundary predictor is trained to capture structural changes in attention patterns rather than task-specific semantics. Empirically, DHSA generalizes to benchmarks with different distributions. The configuration includes a context window size of $w = 4$ (i.e., 8 tokens per position), 8 attention heads, average pooling, an MLP hidden size of 256, and a total model size of 20 MB shared across layers and datasets. We further analyze the sensitivity of boundary-detection and NMS hyperparameters in Table 7. Performance is stable across a broad range of moderate values, and the same hyperparameters are used across model families in our experiments. Additional details for boundary predictor training and backend implementation are provided in Appendices A and C.

*Table 1.* Performance of different methods across base models on LongBench (token density = 12.5%). All baselines are evaluated under matched sparsity settings. DuoAttention, SeerAttention, and Quest do not support Qwen2.5-3B-Instruct or gemma-2-2b-it, and are therefore omitted for those models.

| Methods | Single Doc QA | Multi Doc QA | Summ. | Few-shot Learning | Synth. | Code | Avg. |
|---|---|---|---|---|---|---|---|
| *Llama-3.1-8B-Instruct (4-bit)* | 22.0 | 10.5 | 29.4 | 68.3 | 44.0 | 22.3 | 32.7 |
| StreamingLLM | 15.0 | 6.8 | **27.7** | 62.3 | 26.0 | 22.0 | 27.0 |
| StreamingLLM w/ dilated | 15.1 | 6.7 | 27.6 | 62.3 | 26.0 | 22.1 | 27.0 |
| StreamingLLM w/ strided | 12.0 | 5.0 | 25.5 | 58.9 | 11.3 | **24.1** | 23.4 |
| MInference | 17.6 | 8.2 | 26.7 | 67.8 | 24.3 | 22.1 | 28.4 |
| Block Sparse | 16.3 | 7.4 | 22.3 | 59.7 | 44.2 | 20.5 | 27.9 |
| DuoAttention | 13.0 | 6.5 | 26.4 | 60.9 | 3.5 | 22.5 | 23.3 |
| SeerAttention | 19.9 | 10.4 | 25.6 | 68.7 | 40.1 | 19.5 | 30.8 |
| Quest | **22.4** | **10.7** | 27.1 | 60.3 | 45.2 | 21.4 | 30.9 |
| **DHSA** | 18.3 | 10.1 | 26.9 | **68.8** | 45.7 | 22.7 | **31.8** |
| *Qwen2.5-3B-Instruct* | 12.7 | 6.9 | 25.5 | 66.9 | 20.0 | 19.8 | 26.0 |
| StreamingLLM | 9.2 | 6.0 | **25.7** | 54.8 | 2.5 | 19.8 | 20.7 |
| StreamingLLM w/ dilated | 9.7 | 6.2 | 25.0 | 54.0 | 3.3 | 19.6 | 20.7 |
| StreamingLLM w/ strided | 8.1 | 7.0 | 24.7 | 47.2 | 2.1 | 18.1 | 18.8 |
| MInference | 10.1 | 5.5 | 24.1 | 57.0 | 2.5 | 19.6 | 21.1 |
| Block Sparse | 9.6 | 4.6 | 21.3 | 56.1 | 10.0 | 17.5 | 20.6 |
| **DHSA** | **11.3** | **7.9** | 24.7 | **67.2** | 14.0 | **21.6** | **25.3** |
| *gemma-2-2b-it* | 27.0 | 26.2 | 24.7 | 65.0 | 7.5 | 25.5 | 30.9 |
| StreamingLLM | 13.6 | 15.2 | 22.9 | 53.4 | 7.5 | 26.2 | 23.9 |
| StreamingLLM w/ dilated | 14.8 | 16.5 | 22.3 | 52.2 | **10.0** | **28.7** | 24.7 |
| StreamingLLM w/ strided | 14.1 | 15.1 | **23.2** | 51.2 | **10.0** | 24.4 | 23.7 |
| MInference | 17.3 | 20.8 | 22.3 | 58.0 | 5.0 | 27.5 | 26.3 |
| Block Sparse | 11.4 | 17.4 | 16.3 | 51.6 | 2.5 | 25.1 | 21.6 |
| **DHSA** | **27.6** | **24.5** | 23.1 | **64.4** | 7.5 | 25.4 | **30.3** |

*Table 2.* Comparison of sparse attention methods. We evaluate whether each method (i) accelerates prefill, (ii) is model-agnostic at the algorithm or implementation level, and (iii) is hardware-compatible (HW), i.e., runs on both GPU and CPU. $^\dagger$Conceptually architecture-agnostic, though current open-source implementations support only a subset of LLM families.

| Method | Prefill | Model-Agnostic | HW |
|---|---|---|---|
| StreamingLLM | | ✓ | ✓ |
| MInference | ✓ | ✓ | |
| Block-Sparse | ✓ | | |
| DuoAttention | | ✓$^\dagger$ | |
| SeerAttention | ✓ | | |
| Quest | | ✓$^\dagger$ | |
| **DHSA** | ✓ | ✓ | ✓ |

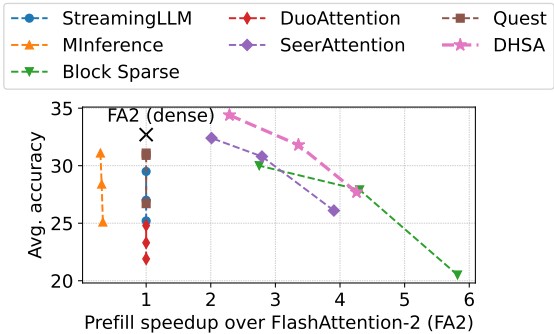

*Figure 5.* Accuracy and (kernel-level) prefill speedup trade-off on LongBench with LLaMA-3.1-8B (4-bit). The results are measured on a single RTX 3090 with batch size 1, using token densities of 6.25%, 12.5%, and 25.0%.

**Baselines.** In addition to dense attention, we include eight sparse attention baselines that keep the LLM backbone frozen: StreamingLLM and its dilated and strided variants (Xiao et al., 2024; Beltagy et al., 2020; Child, 2019), MInference (Jiang et al., 2024), Block-Sparse Attention (Guo et al., 2024), DuoAttention (Xiao et al., 2025), SeerAttention (Gao et al., 2024) and Quest (Tang et al., 2024). Following prior prefill-acceleration work, prefill-oriented methods, including MInference, Block-Sparse Attention, SeerAttention, and DHSA, use sparse attention during prefill and standard dense attention during decoding. For decoding-oriented methods, including DuoAttention and

Quest, we use their original implementations with dense prefill and sparse decoding, and report them as reference baselines for downstream accuracy. Detailed configurations and hyperparameters for all baselines are provided in Appendix C.

**LongBench.** As shown in Table 1, DHSA achieves the best overall performance on LongBench compared to all baseline methods. Notably, DHSA matches the performance of the original dense attention baseline with a token density of only 12.5% (defined in Appendix B.6), while providing significant reductions in latency and memory usage.

*Table 3.* Effect of Token Density on Method Performance (LongBench).

| Methods | Single Doc QA | Multi Doc QA | Summ. | Few-shot Learning | Synth. | Code | Avg. |
|---|---|---|---|---|---|---|---|
| *Llama-3.1-8B-Instruct (4-bit)* | 22.0 | 10.5 | 29.4 | 68.3 | 44.0 | 22.3 | 32.7 |
| *Density = 6.25%* | | | | | | | |
| StreamingLLM | 13.1 | 5.3 | **26.4** | 60.4 | 20.2 | **23.9** | 25.2 |
| MInference | 14.4 | 6.9 | 20.9 | **64.4** | 20.0 | 19.7 | 25.1 |
| Block Sparse | 10.6 | 6.0 | 19.2 | 46.7 | 22.3 | 17.6 | 20.5 |
| DuoAttention | 11.5 | 5.2 | 24.8 | 59.5 | 2.7 | 21.0 | 21.9 |
| SeerAttention | 20.0 | 8.0 | 23.9 | 59.4 | 24.0 | 17.4 | 26.1 |
| Quest | **20.4** | **10.2** | 25.3 | 58.3 | 24.0 | 18.5 | 26.7 |
| **DHSA** | 18.1 | 7.2 | 21.2 | **64.4** | 36.3 | 20.2 | **27.7** |
| *Density = 12.5%* | | | | | | | |
| StreamingLLM | 15.0 | 6.8 | **27.7** | 62.3 | 26.0 | 22.0 | 27.0 |
| MInference | 17.6 | 8.2 | 26.7 | 67.8 | 24.3 | 22.1 | 28.4 |
| Block Sparse | 16.3 | 7.4 | 22.3 | 59.7 | 44.2 | 20.5 | 27.9 |
| DuoAttention | 13.0 | 6.5 | 26.4 | 60.9 | 3.5 | 22.5 | 23.3 |
| SeerAttention | 19.9 | 10.4 | 25.6 | 68.7 | 40.1 | 19.5 | 30.8 |
| Quest | **22.4** | **10.7** | 27.1 | 60.3 | 45.2 | 21.4 | 30.9 |
| **DHSA** | 18.3 | 10.1 | 26.9 | **68.8** | 45.7 | **22.7** | **31.8** |
| *Density = 25.0%* | | | | | | | |
| StreamingLLM | 20.5 | 7.7 | 28.2 | 61.6 | 36.3 | 22.8 | 29.5 |
| MInference | 21.9 | 9.5 | 28.8 | 67.5 | 35.9 | 21.5 | 31.1 |
| Block Sparse | 18.2 | 8.9 | 26.7 | 62.4 | 42.1 | 23.2 | 30.0 |
| DuoAttention | 14.5 | 8.0 | 28.1 | 62.4 | 5.0 | **23.8** | 24.8 |
| SeerAttention | 21.7 | 9.2 | 28.2 | 68.7 | 44.5 | 23.2 | 32.4 |
| Quest | 22.8 | **10.8** | 28.7 | 60.5 | 41.2 | 23.6 | 31.1 |
| **DHSA** | **24.3** | 10.2 | **28.9** | **69.4** | **52.7** | 23.7 | **34.4** |

*Figure 6.* DHSA speedup over FA2 at the attention kernel level during the prefill stage. Results are obtained using 4-bit Llama-3.1-8B-Instruct on an NVIDIA RTX 3090. DHSA consistently reduces latency across all evaluated sequence lengths and density settings.

**Effect of Token Density.** As shown in Table 3, we investigate the impact of token density (defined in Appendix B.6) on model performance. Across all density levels, DHSA consistently outperforms baseline methods and benefits most from larger attention budgets. Overall, DHSA scales monotonically with density and maintains stable performance across tasks.

**Needle-In-A-Haystack.** As shown in Figure 1a, our method reliably retrieves information placed at different positions across context windows ranging from 1K to 100K tokens. In contrast, baselines such as Block Sparse Attention, while effective in reducing latency, suffer a sharp performance drop once the critical information lies outside their restricted attention ranges (see Figure 13).

**RULER.** To complement Needle-In-A-Haystack (NIAH) with a benchmark that explicitly controls context length and includes reasoning-oriented tasks beyond simple retrieval, we further evaluate DHSA on RULER (Hsieh et al., 2024). We report overall performance across all RULER tasks and additionally analyze `QA-1`, `QA-2`, and `VT`, which require combining information from multiple distributed spans and are therefore more challenging than single-needle retrieval. As shown in Table 5, DHSA achieves strong overall performance under the same token-density setting, obtaining the best average among sparse-attention baselines at longer contexts, including 32K and 48K, while remaining competitive at shorter lengths. Task-level results are provided in Table 6, further confirming that DHSA maintains robust performance across those distributed-information tasks. These results suggest that DHSA can preserve multiple relevant regions beyond simple retrieval-style locality, without exhibiting an abrupt accuracy collapse as context length increases.

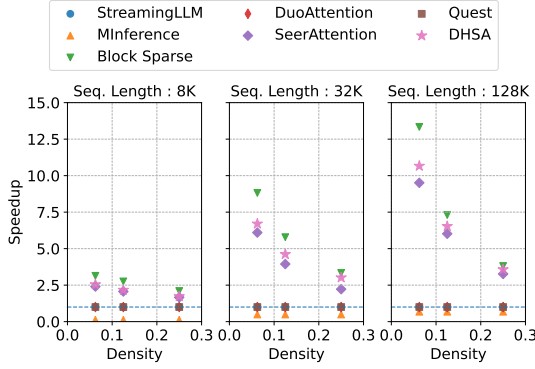

*Figure 7.* Kernel-level prefill speedup over Flash Attention-2 across multiple sparse attention methods. Note that StreamingLLM, DuoAttention, and Quest primarily target KV-cache compression and therefore do not accelerate prefill latency.

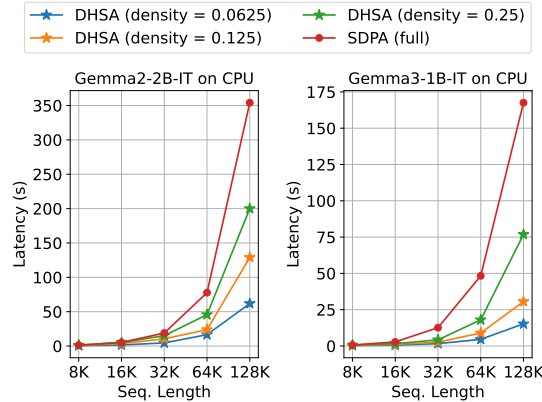

*Figure 8.* DHSA speedup over Torch.SDPA at the attention-kernel level during the prefill stage. Measurements are obtained on an Intel Core 5 120U CPU.

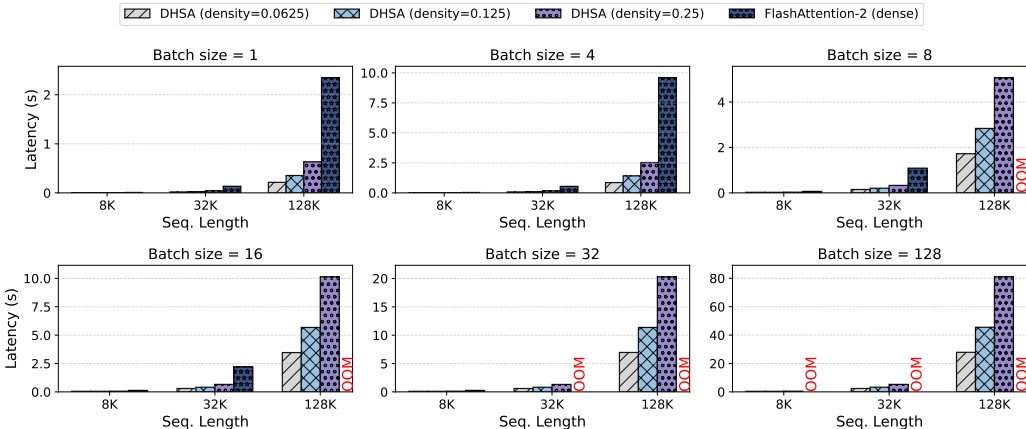

*Figure 9.* DHSA prefill speedup over FlashAttention-2 at the kernel level across different batch sizes. DHSA processes batches with a simple for-loop, which, under memory-constrained settings, is faster than batched FA2 and avoids the OOM failures that FA2 often encounters. Results are measured with 4-bit Llama-3.1-8B-Instruct on an NVIDIA RTX 3090.

**Latency Comparison.** As shown in Figure 6, DHSA consistently reduces prefill latency over FlashAttention-2 across sequence lengths and density levels. Compared with other sparse baselines (Figure 7), DHSA achieves substantial acceleration while approaching the speed of Block-Sparse Attention. These gains generalize beyond GPUs, as DHSA also accelerates Torch.SDPA on CPU (Figure 8). We also show TTFT results on LLaMA-3.1-8B (4-bit) in Figure 1b. Additional TTFT results on LongBench and an overhead breakdown are provided in Tables 9 and 10; DHSA substantially reduces the average TTFT over LongBench, and its routing overhead is outweighed by the savings from sparse attention at longer context lengths. All reported latency results focus on prefill latency or TTFT, consistent with DHSA's target setting of prefill acceleration.

**Batch Inference.** Batch inference is inherently challenging for dynamic sparse attention. Rather than enforcing a shared sparsity mask across all sequences in a batch, DHSA processes each example sequentially with a lightweight for-loop. Under **memory-constrained** settings, this strategy is faster than batched FA2 and avoids the OOM failures that FA2 frequently encounters (Figure 9).

**Ablation Study on the Components of DHSA.** The results in Table 4 show that dynamic chunking substantially improves performance by better capturing the semantic structure of tokens. Furthermore, robust chunk representation provides additional gains by preventing important tokens from being overshadowed by less relevant ones. Detailed analysis is provided in Appendix D.

## 5. Related Work

**Sparse Attention.** The quadratic cost of attention has motivated sparse alternatives, broadly divided into *static*

*Table 4.* Performance of different ablation methods using Llama-3.1-8B (4-bit) on LongBench (density = 12.5%).

| Methods | Single Doc QA | Multi Doc QA | Summ. | Few-shot Learning | Synth. | Code | Avg. |
|---|---|---|---|---|---|---|---|
| DHSA | **18.3** | **10.1** | 26.9 | **68.8** | **45.7** | **22.7** | **31.8** |
| DHSA w/o robust chunk repr. | 17.1 | 8.2 | **27.0** | 67.2 | 45.0 | 21.4 | 30.7 |
| DHSA w/o dynamic chunking | 16.4 | 7.4 | 22.1 | 59.8 | 44.0 | 20.6 | 28.0 |
| DHSA w/o robust chunk repr. & dynamic chunking | 16.3 | 7.4 | 22.3 | 59.7 | 44.2 | 20.5 | 27.9 |

and *dynamic* methods. *Static* patterns such as sliding windows (Child, 2019), dilated/strided schemes (Beltagy et al., 2020; Ding et al., 2023), and local–global mixtures (Beltagy et al., 2020; Zaheer et al., 2020) are usually hardwired at pretraining, making them difficult to use as drop-in replacements without accuracy loss. *Dynamic* methods adapt sparsity at inference. Examples include MInference (Jiang et al., 2024), which combines pre-defined templates with kernel-aware indexing, and block-sparse attention (Guo et al., 2024), which uses blockwise attention to select important blocks. However, these approaches still rely heavily on templates or heuristics, limiting their ability to capture semantic structure.

**Long-Context Modeling.** Long-context modeling faces both high attention cost and large KV cache storage. Prefill optimizations include state-space models (Gu et al., 2022; Gu & Dao, 2023), linear attention (Sun et al., 2023), memory-based methods (Munkhdalai et al., 2024; Yang et al., 2024b), hybrids (Lieber et al., 2024), and prompt compression (Li et al., 2023), though most require retraining the LLM backbone. Recent work (Jiang et al., 2024; Guo et al., 2024) instead uses predefined templates or heuristics. Decoding optimizations focus on static (Xiao et al., 2024; Han et al., 2024) and dynamic (Zhang et al., 2023; Liu et al., 2023) KV cache dropping, and KV cache quantization (Liu et al., 2024), but these do not reduce prefill attention cost.

**Memory-Constrained LLM Inference.** The primary challenges for memory-constrained long-context inference are the memory footprint and compute cost. Practical systems address these constraints by combining model compression (8/4-bit quantization (Dettmers et al., 2023), activation/KV quantization (Liu et al., 2024), pruning (Frantar & Alistarh, 2023)), capacity transfer (distillation to small models (Sanh et al., 2019; Zhang et al., 2024)), kernel-level optimizations (Flash/SDPA attention (Dao et al., 2022), operator fusion, paged attention (Kwon et al., 2023), speculative decoding (Leviathan et al., 2023)), and context management (prompt compression (Li et al., 2023), cache eviction/dropping (Xiao et al., 2024; Zhang et al., 2023; Li et al., 2024)). These techniques are complementary: quantization reduces storage and compute costs, KV compression and eviction bound runtime memory, while decoding accelerations reduce token-generation latency. Our method,

*Table 5.* Performance of different methods on RULER (token density = 12.5%). All baselines are evaluated under matched sparsity settings. We extend the controlled-length evaluation to 48K tokens, the longest context length for which dense attention remains feasible in our setting.

| Method | 4K | 8K | 16K | 32K | 48K |
|---|---|---|---|---|---|
| *Llama-3.1-8B-Instruct (4-bit)* | 95.8 | 93.8 | 93.2 | 81.5 | 75.2 |
| MInference | 91.7 | 88.4 | 88.6 | 74.7 | 64.2 |
| BlockSparse | 60.3 | 65.4 | 69.3 | 53.0 | 43.1 |
| DuoAttention | 92.0 | **89.3** | **89.7** | 72.9 | 69.1 |
| SeerAttention | 86.9 | 89.0 | 89.1 | 75.0 | 53.6 |
| Quest | 87.7 | 86.0 | 87.2 | 73.9 | 68.2 |
| **DHSA** | **92.1** | 88.5 | 88.7 | **76.2** | **71.5** |

introducing input-adaptive sparsity to cut prefill attention overhead while preserving accuracy, can be integrated as a drop-in component within existing stacks.

## 6. Conclusion

We presented Dynamic Hierarchical Sparse Attention (DHSA), which integrates dynamic chunking and hierarchical sparsity prediction for efficient long-context attention. Experiments on the Needle-in-a-Haystack test, LongBench and RULER demonstrate that DHSA achieves comparable accuracy to dense attention while significantly reducing latency. By adapting to input-dependent attention patterns, DHSA provides a practical and efficient solution for long-context modeling under memory-constrained settings.

## Impact Statement

This paper presents work whose goal is to advance the field of Machine Learning. There are many potential societal consequences of our work, none which we feel must be specifically highlighted here.

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

# A. Boundary Detection

## A.1. Problem Formulation

We formulate the chunking task as a **boundary detection** problem, where the objective is to determine whether each token position marks the *end of a chunk*. Formally, for each position $i \in [0, L-1]$, we define a boundary indicator function

$$\delta(i) = \begin{cases} 1 & \text{if } i = b_k \text{ for some } k, \\ 0 & \text{otherwise,} \end{cases}$$

indicating that the token at position $i$ is the last token of a chunk. This end-boundary prediction approach aligns naturally with the way sequences are segmented, as it allows the model to determine when a coherent segment of context has concluded. By framing the task this way, we can leverage binary classification methods to adaptively segment sequences in DHSA.

This probability, denoted as $\Pr(\delta(i) = 1)$, is predicted based on the local key representations surrounding $i$, using a boundary prediction function that takes as input two windows centered at $i$:

$$[\mathbf{k}_{i-w+1}, \cdots, \mathbf{k}_i], \quad [\mathbf{k}_{i+1}, \cdots, \mathbf{k}_{i+w}],$$

where $w$ is a hyperparameter that determines the receptive field and $\mathbf{k}_j$ denotes the key vector corresponding to token $j$.

Local context is sufficient for boundary prediction because chunk boundaries are determined by **local changes in semantic similarity**. Intuitively, if the left window (preceding tokens) and the right window (succeeding tokens) are highly similar, the two regions likely belong to the same chunk, and no boundary should be placed. Conversely, a sharp drop in similarity between these windows indicates a topic or context shift, suggesting the end of a chunk. Moreover, focusing on local context rather than the full sequence significantly reduces computation. Evaluating boundaries requires processing only tokens per position which is essential for **efficiency**.

## A.2. Architecture

We define the boundary prediction function as a neural network. After exploring various architectures and hyperparameters, we finalize the design shown in Figure 4. This architecture was chosen because it strikes a balance between **expressiveness**, **efficiency**, and **robustness**. The encoder effectively captures contextualized token embeddings for both the left and right windows. The feature fusion module proved more stable and discriminative than using a single metric in isolation. The final MLP is shallow enough to maintain low latency but still has sufficient capacity. By keeping the receptive field limited to $2w$ tokens, the total prediction cost is linear to token length $L$.

**Encoder.**  Our encoder consists of a lightweight self-attention block followed by pooling. Given a window of key vectors $\mathbf{K} \in \mathbb{R}^{w \times d}$, we compute

$$\mathbf{Q}^b = \mathbf{K}W_Q^b, \quad \mathbf{K}^b = \mathbf{K}W_K^b, \quad \mathbf{V}^b = \mathbf{K}W_V^b,$$

where $W_Q^b, W_K^b, W_V^b$ are learned projection matrices. We then apply standard multi-head self-attention and aggregate the resulting token-level representations using pooling to obtain a fixed-length vector. The encoder parameters are independent of the base LLM. We use average pooling to obtain the context vector, as it has proven to be sufficiently reliable. Although we also tested learnable pooling, average pooling provided satisfactory results in our experiments.

**Feature Fusion.**  In the feature fusion module, we combined the raw context vectors $\mathbf{k}_{\text{left}}, \mathbf{k}_{\text{right}}$, absolute differences $|\mathbf{k}_{\text{left}} - \mathbf{k}_{\text{right}}|$, multiplicative interactions $\mathbf{k}_{\text{left}} \odot \mathbf{k}_{\text{right}}$, and cosine similarity $\text{sim}(\mathbf{k}_{\text{left}}, \mathbf{k}_{\text{right}})$ because each signal captures complementary aspects of boundary semantics. The raw vectors preserve local context information, absolute differences highlight directional changes between left and right spans, multiplicative interactions emphasize co-activation patterns, and cosine similarity provides a normalized measure of alignment. Together, these features make the boundary predictor more robust to scale, length, and semantic variation. In ablations, we found that using only a single similarity measure (e.g., cosine similarity) was less stable, whereas combining multiple signals yielded consistently better boundary detection.

## A.3. Automatic Labeling

To train the boundary predictor, we automatically derive ground-truth labels from attention scores, avoiding the need for manual annotation. We adopt this intermediate labelling strategy instead of end-to-end training from final task performance.

End-to-end optimization is computationally intractable, as it would require differentiating through boundary indices with only sparse, delayed supervision from downstream metrics. In contrast, derived labels provide dense, local supervision, turning boundary detection into a well-defined, efficient classification problem that still captures the structural cues implicit in the model's own attention behavior.

Specifically, we analyze accumulated attention mass patterns. Tokens within a coherent span typically exhibit consistent accumulated attention profiles, meaning the distribution of attention mass over preceding tokens remains relatively stable across positions within the span. In contrast, a boundary is often marked by a sudden change in this profile, such as when the subsequent token's accumulated attention shifts sharply toward a different subset of preceding tokens.

Figure 10 illustrates our automatic labeling strategy. For each candidate position, we examine the accumulated attention mass patterns in its left window and right window, where each row in the heatmap corresponds to a token and color intensity indicates the magnitude of accumulated attention mass. In the left example, the left and right windows (both outlined in blue) exhibit highly similar attention profiles, indicating that the tokens around this position belong to the same coherent span; thus, the position is not labeled as a boundary. In the right example, the attention profiles of the left window (blue) and right window (green) differ markedly, signaling a sharp change in attention behavior. This difference suggests that the position lies at the end of a chunk and should be labeled as a boundary.

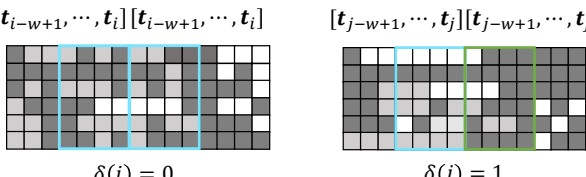

$\delta(i) = 0$      $\delta(j) = 1$

*(a)* Non-boundary: similar left/right attention mass.
*(b)* Boundary: different left/right attention mass.

*Figure 10.* Illustration of our automatic labeling strategy. The core intuition is that tokens within a coherent span tend to receive similar attention distributions. Positions where the left and right windows show similar attention distributions are labeled as non-boundary, whereas positions with sharp differences between the two windows indicate a boundary.

Formally, let the token sequence be

$$\mathbf{T} = [\mathbf{t}_0, \mathbf{t}_1, \ldots, \mathbf{t}_{L-1}]$$

with length $L$, and let $A \in \mathbb{R}^{L \times L}$ denote its attention matrix, where $A_{u,v}$ is the attention weight from token $u$ to token $v$.

For each position $i \in [0, L-1]$, we consider whether it marks the end of a chunk. To do so, we examine two local windows of size $w$ on either side of $i$, i.e., the past window: tokens $\{\mathbf{t}_{i-w+1}, \ldots, \mathbf{t}_i\}$ and the future window: tokens $\{\mathbf{t}_{i+1}, \ldots, \mathbf{t}_{i+w}\}$.

We then compute the past cumulative attention mass:

$$a_{\text{past}}(i) = \frac{1}{L-1-i-w} \sum_{u=i+w+1}^{L-1} \sum_{v=i-w+1}^{i} A_{u,v}, \tag{5}$$

and the future cumulative attention mass:

$$a_{\text{fut}}(i) = \frac{1}{L-1-i-w} \sum_{u=i+w+1}^{L-1} \sum_{v=i+1}^{i+w} A_{u,v}, \tag{6}$$

where $w = 4$ is the local window size (receptive field), and the outer sum index $u$ iterates over future tokens (beyond $i+w$) while the inner sum index $v$ iterates over the tokens in the corresponding window.

We define the attention ratio:

$$r_i = \frac{\max\left(a_{\text{fut}}(i), a_{\text{past}}(i)\right) + \varepsilon}{\min\left(a_{\text{fut}}(i), a_{\text{past}}(i)\right) + \varepsilon}, \tag{7}$$

where $\varepsilon = 0.001$ is added for numerical stability.

Given the maximum allowed number of chunks $N_c$ and a threshold $\theta_r = 1.1$, we select the top $N_c - 1$ positions $i$ whose $r_i$ exceeds $\theta_r$ (with positions 0 and $L$ always included as boundaries), with $N_c$ serving as the primary constraint in practice.

## A.4. Training of the Boundary Predictor

We describe below the key implementation details of our training pipeline that are non-trivial and contribute meaningfully to performance.

**Soft Labels.** Instead of using hard labels, which are obtained by selecting the top $N_c - 1$ positions with the highest attention ratios, we adopt a soft labeling strategy. Hard labels are inherently sensitive to the chunk number constraint, i.e., the fixed number $N_c - 1$ of chunk boundaries to be selected per sequence. Under this scheme, a position with a given ratio $r$ might receive a label of 1 for one choice of $N_c$ but a label of 0 for another, even though its underlying ratio has not changed. This sensitivity can confuse the model and discard useful information about positions near the cutoff.

To address this, we convert the attention ratio $r$ into a continuous probability value using the following transformation:

$$p = \sigma\big(\alpha \cdot (\log(r + \zeta) - \beta)\big) \tag{8}$$

where $r$ is the ratio, $\zeta = 10^{-6}$ is a small constant for numerical stability, $\alpha = 2.0$ and $\beta = \lg(2.0)$ are scalar parameters controlling the slope and offset, with base $e \approx 2.71828$, and $\sigma(x) = \frac{1}{1+e^{-x}}$ is the sigmoid function.

This maps the ratio to a probability in the $[0, 1]$ range, preserving the relative ordering of positions and capturing confidence without enforcing a hard cutoff. Positions with higher ratios produce probabilities closer to 1, but those with moderately high ratios still receive meaningful supervision. This approach is in line with knowledge distillation, where soft targets have been shown to improve generalization by providing richer learning signals than binary labels.

**Loss.** Using the conventional Binary Cross-Entropy (BCE) loss for training our boundary predictor in dynamic chunking, we identify two key issues: (1) *class imbalance*, where boundary tokens are much fewer than non-boundary tokens, and (2) *varying sample difficulty*, where some boundaries are easier to detect than others. To address these, we adopt the **focal BCE loss**, which down-weights the contribution of easy examples and amplifies the focus on harder, misclassified cases. The weighting term $(1 - p_i)^\gamma$ automatically reduces the loss for well-classified positions (large $p_i$ for positives, small $p_i$ for negatives), while the fixed positive-class weight $w$ offsets the imbalance between boundary and non-boundary positions.

$$\mathcal{L}_i = (1 - p_i)^\gamma \Big[ -wy_i \log p_i - (1 - y_i) \log(1 - p_i) \Big] \tag{9}$$

where $y_i \in [0, 1]$ is the ground-truth soft label for position $i$, $z_i \in \mathbb{R}$ is the logit produced by the boundary predictor, $p_i = \sigma(z_i) = \frac{1}{1+e^{-z_i}}$ is the predicted probability, $w = 1.3$ is the fixed positive-class weight for class imbalance, and $\gamma = 2.0$ is the focal parameter controlling emphasis on hard examples.

**Data Preparation.** We analyze existing long-context datasets for both training and inference. For training, we select Long Data Collections[1], trivia QA (Joshi et al., 2017), ChatQA2 (Xu et al., 2025) that mainly focus on high-quality question answering and summarization tasks. We inspected the datasets and found that the samples are quite similar. To accelerate training, we apply sampling by selecting the first 10,000 samples from each dataset for training. For evaluation, we use the first 100 samples from the validation set of each dataset.

**Metrics.** We monitor the following metrics during training to facilitate debugging and to compare different hyperparameter configurations. We track the training loss across all layers (results shown for Gemma2-2b-it, which has 26 layers in total). For training, we evaluate precision, recall, and F1 score for the positive class (where soft labels $\geq 0.5$ are treated as positive), as well as top-$K$ overlap with $K = 500$. Top-$K$ overlap is defined as the number of overlapping positions divided by $K$, and we monitor this metric because, during inference, the top positions are selected as boundaries. For validation, we compute precision, recall, F1 score, and top-$K$ overlap ($K = 500$) on the validation set.

**Acceleration.** We further accelerate training through the following strategies. (1) *Storing only boundary labels*: Instead of storing both input embeddings and boundary labels, which would consume excessive memory, we store only the labels. This allows the model to perform dynamic chunking directly from the loaded labels without recalculating them. Additionally, the labels can be generated in parallel for all training samples, significantly improving efficiency. (2) *Training all layers simultaneously*: The boundary predictor needs to be trained for all layers. Rather than performing multiple forward passes, one for each layer, we perform a single forward pass of the language model per sample to obtain input embeddings for all layers. These embeddings are then used to train the predictor for all layers simultaneously.

**Training Cost.** The boundary predictor is trained separately for each backbone family while keeping the corresponding LLM backbone frozen. In our implementation, we train the predictor for 10,000 steps on a single NVIDIA RTX 3090 GPU

---

[1] https://huggingface.co/datasets/togethercomputer/Long-Data-Collections

(24 GB). For Gemma-2-2B-it, training takes 3.20 seconds per iteration, corresponding to approximately 8.89 hours in total. Since the predictor has only 20 MB of parameters and is shared across layers and datasets within each backbone family, this cost is a one-time lightweight calibration cost rather than LLM finetuning.

### A.5. Inference of the Boundary Predictor

We observe that selecting positions purely based on the topK scores can lead to suboptimal boundaries due to noise and closely spaced high-score peaks. To mitigate this, we adopt Non-Maximum Suppression (NMS) (Neubeck & Van Gool, 2006), a technique widely used for eliminating redundant detections in computer vision. The core idea is to keep only the highest-scoring candidate within a local neighborhood, thereby producing well-separated, reliable boundaries.

We first obtain the boundary scores by computing the boundary predictor's output scores for all positions in the sequence, which represent the model's confidence that a given position marks a chunk end. Next, we identify candidate boundary positions by selecting all positions whose scores exceed a minimal confidence threshold, thereby pruning out low-probability positions while retaining multiple plausible candidates. The remaining candidates are then sorted in descending order of their scores, ensuring that higher-confidence positions are considered first during suppression. We subsequently apply NMS: starting with the highest-scoring candidate, we mark it as a boundary and remove any other candidates within a specified window size (e.g., 8 or 64 tokens) of this position, as they are considered overlapping or too close to be separate boundaries. After suppression, the final set consists of well-spaced local maxima that are more robust to noise and score fluctuations.

Applying NMS with a small window size (e.g., 8) yields better results than using no NMS. For tasks requiring broader context segmentation, a larger NMS window size (e.g., 64) further improves performance. In some cases, we also augment the output with explicit boundaries such as \n, \n\n, or prior information from structured prompts to better align with semantic structure.

## B. Theoretical Analysis

In this section, we provide a theoretical analysis that explains why *Dynamic Hierarchical Sparse Attention* (DHSA) can achieve higher attention recall than fixed block-sparse attention under the same sparsity budget.

### B.1. Setup and Notation

We consider a single attention head and a single layer, and start with a single query token $i \in \{1, \ldots, L\}$. Let

$$a_{ij} \geq 0, \quad j = 1, \ldots, L$$

denote the (unnormalized) attention scores from query $i$ to keys $j$. Let $I_i^\star \subseteq \{1, \ldots, L\}$ be the *oracle important set* for query $i$ (e.g., the indices of the top-$K$ keys by $a_{ij}$), and define the corresponding binary mask row

$$M_i^\star(j) = \mathbf{1}\{j \in I_i^\star\}.$$

Given any approximate sparsity pattern $\widehat{M}_i(j) \in \{0, 1\}$ for query $i$, we measure its *per-row recall* with respect to $M_i^\star$ as

$$\text{Recall}_i(\widehat{M}) = \frac{\sum_{j=1}^L M_i^\star(j)\, \widehat{M}_i(j)}{\sum_{j=1}^L M_i^\star(j)} = \frac{|I_i^\star \cap \widehat{I}_i|}{|I_i^\star|}, \tag{10}$$

where $\widehat{I}_i = \{j : \widehat{M}_i(j) = 1\}$ is the set of keys selected by the approximate method.

We assume that all methods are constrained to use the *same token budget* per query:

$$|\widehat{I}_i| = |I_i^\star| = K. \tag{11}$$

At the matrix level, the overall recall is the average $\text{Recall}(\widehat{M}) = \frac{1}{L} \sum_i \text{Recall}_i(\widehat{M})$. It therefore suffices to compare $\text{Recall}_i$ for a fixed query $i$.

We denote by $M_i^{\text{DHSA}}$ and $M_i^{\text{block}}$ the binary masks produced by our method (DHSA) and by fixed block-sparse attention, respectively, and compare $\text{Recall}_i(M^{\text{DHSA}})$ to $\text{Recall}_i(M^{\text{block}})$ under the same budget equation 11.

## B.2. Semantic Segment Assumption

We formalize the intuition that important keys for a given query tend to cluster into contiguous "semantic segments" (e.g., phrases, sentences, paragraphs).

**Definition B.1** (Semantic segments). Let $\{1, \ldots, L\}$ be partitioned into contiguous segments

$$0 = b_0 < b_1 < \cdots < b_C = L,$$

and define segment $c$ as the index set

$$\mathcal{S}_c = \{b_{c-1} + 1, \ldots, b_c\}, \quad c = 1, \ldots, C.$$

For a fixed query $i$, define the *segment mean importance*

$$\mu_{ic} = \frac{1}{|\mathcal{S}_c|} \sum_{j \in \mathcal{S}_c} a_{ij}. \tag{12}$$

We assume that scores vary slowly within segments but can change substantially across segments.

**Assumption B.2** (Intra-segment homogeneity and inter-segment separation). Define

$$\epsilon_{\text{intra}} = \max_{i,c} \max_{j,j' \in \mathcal{S}_c} |a_{ij} - a_{ij'}|$$

and

$$\Delta_{\text{inter}} = \min_i \min_{c \neq c'} |\mu_{ic} - \mu_{ic'}|.$$

We assume

$$\epsilon_{\text{intra}} \ll \Delta_{\text{inter}}. \tag{13}$$

Assumption B.2 captures a regime where tokens within the same semantic segment have similar importance, while large changes occur when moving between different segments (e.g., across sentence or topic boundaries).

## B.3. Relevant vs. Irrelevant Segments

Fix a query $i$. Suppose:

1. All segments have equal length $m$: $|\mathcal{S}_c| = m$ for all $c$, hence $C_m = L$.

2. Each segment is either *relevant* or *irrelevant* for query $i$:
    - If segment $c$ is relevant, then all tokens in $\mathcal{S}_c$ are important: $M_i^\star(j) = 1$ for all $j \in \mathcal{S}_c$.
    - If segment $c$ is irrelevant, then all tokens in $\mathcal{S}_c$ are unimportant: $M_i^\star(j) = 0$ for all $j \in \mathcal{S}_c$.

Let $\mathcal{C}_{\text{rel}} \subseteq \{1, \ldots, C\}$ be the set of relevant segments for query $i$, with size $|\mathcal{C}_{\text{rel}}| = R$. By construction, the oracle important set is

$$I_i^\star = \bigcup_{c \in \mathcal{C}_{\text{rel}}} \mathcal{S}_c, \quad |I_i^\star| = R_m. \tag{14}$$

We consider a per-row sparsity budget equal to the number of truly important tokens:

$$K = |I_i^\star| = R_m. \tag{15}$$

Under this budget, the oracle can achieve $\text{Recall}_i(M^\star) = 1$.

## B.4. DHSA with Dynamic Chunking

DHSA predicts *chunk boundaries* $\hat{b}_0, \ldots, \hat{b}_{\hat{C}}$ and induced chunks

$$\hat{\mathcal{S}}_k = \{\hat{b}_{k-1} + 1, \ldots, \hat{b}_k\}, \quad k = 1, \ldots, \hat{C}.$$

For each chunk and query $i$, DHSA computes a *length-normalized chunk score*

$$\hat{\mu}_{ik} = \frac{1}{|\hat{\mathcal{S}}_k|} \sum_{j \in \hat{\mathcal{S}}_k} a_{ij}, \tag{16}$$

implemented via robust chunk representation. DHSA selects the top-$N_b$ chunks according to $\hat{\mu}_{ik}$ and then selects tokens within these chunks until the budget $K$ is filled.

To relate chunks to the true segments $\mathcal{S}_c$, we assume the boundary predictor is reasonably accurate.

**Assumption B.3** (Boundary accuracy). For each true segment $\mathcal{S}_c$, the predicted boundaries may shift the true endpoints by at most $\delta$ tokens on each side. Concretely: each $\mathcal{S}_c$ is covered by the union of at most two predicted chunks, and the number of tokens of $\mathcal{S}_c$ that lie in chunks dominated by other segments is at most $2\delta$:

$$|\mathcal{S}_c \setminus \hat{\mathcal{S}}_c| \le 2\delta, \tag{17}$$

for some appropriate correspondence between true segments and predicted chunks.

Combined with Assumption B.2, this implies that chunk means preserve the ranking of segment means.

**Lemma B.4** (Chunk score stability). *Under Assumptions B.2 and B.3, each predicted chunk $\hat{\mathcal{S}}_k$ overlaps primarily with some true segment $\mathcal{S}_c$, and its mean $\hat{\mu}_{ik}$ satisfies*

$$|\hat{\mu}_{ik} - \mu_{ic}| \le O(\epsilon_{\text{intra}}) + O\left(\frac{\delta}{m} \Delta_{\text{inter}}\right).$$

*In particular, if $\epsilon_{\text{intra}}$ and $\delta/m$ are sufficiently small, then the ranking of chunk means $\hat{\mu}_{ik}$ is identical to the ranking of segment means $\mu_{ic}$ for query $i$.*

*Proof sketch.* Each chunk $\hat{\mathcal{S}}_k$ is a mixture of tokens from at most two adjacent true segments (by Assumption B.3), and the fraction of misassigned tokens from neighboring segments is bounded by $\delta/m$. The mean $\hat{\mu}_{ik}$ is therefore a convex combination of the means of at most two true segments plus intra-segment noise bounded by $\epsilon_{\text{intra}}$. Using equation 13 to compare this convex combination to the dominant segment mean $\mu_{ic}$ yields the desired bound and ranking preservation. □

We can now bound the recall of DHSA in the above model.

**Proposition B.5** (Near-oracle recall of DHSA). *Consider the model above with segment length $m$, relevant segments $\mathcal{C}_{\text{rel}}$, and budget $K = R_m$ as in equation 15. Suppose Assumptions B.2 and B.3 hold and the ranking of chunk means matches the ranking of segment means (Lemma B.4). Then there exists a choice of $N_b$ and token-level selection within the chosen chunks such that DHSA's per-row recall satisfies*

$$\text{Recall}_i(M^{\text{DHSA}}) \ge 1 - \frac{2\delta}{m}. \tag{18}$$

*Proof.* Since chunk ranking matches segment ranking, DHSA can select exactly the chunks corresponding to all relevant segments $\mathcal{S}_c, c \in \mathcal{C}_{\text{rel}}$, plus possibly some chunks corresponding to irrelevant segments if needed to fill the token budget.

By Assumption B.3, for each relevant segment $\mathcal{S}_c$, at most $2\delta$ tokens lie outside the dominant chunk and may be dropped in the upsampling step. Thus the total number of important tokens missed by DHSA is at most

$$|I_i^\star \setminus I_i^{\text{DHSA}}| \le 2\delta \cdot R.$$

Using $|I_i^\star| = R_m$ from equation 14, the recall equation 10 is

$$\text{Recall}_i(M^{\text{DHSA}}) = 1 - \frac{|I_i^\star \setminus I_i^{\text{DHSA}}|}{|I_i^\star|} \ge 1 - \frac{2\delta R}{R_m} = 1 - \frac{2\delta}{m},$$

which proves equation 18. □

Thus, when $\delta \ll m$, DHSA achieves recall close to 1, i.e., nearly matches the oracle top-$K$ selector under the same sparsity budget.

### B.5. Block-Sparse Attention

We now consider a fixed block-sparse pattern with block size $B$ that partitions the sequence into blocks

$$0 = q_0 < q_1 < \cdots < q_Q = L, \quad q_r = rB,$$

and

$$\mathcal{B}_r = \{q_{r-1} + 1, \ldots, q_r\}, \quad |\mathcal{B}_r| = B.$$

A block-sparse mask selects a subset of blocks for each query and allows attention only within those blocks. We assume that selecting a block contributes all $B$ of its tokens to the budget.

We consider a worst-case but realistic alignment between segments and blocks: each relevant segment of length $m \le B$ is split across two blocks, each containing roughly $m/2$ important tokens and $B - m/2$ unimportant tokens.

Under the same budget $K = R_m$, we show that block-sparse recall can be substantially smaller than DHSA.

**Proposition B.6** (Upper bound on block-sparse recall)**.** *With the worst-case alignment of segments and blocks as described above, block size $B \ge m$, and budget $K = R_m$, the per-row recall of any fixed block-sparse pattern satisfies*

$$\mathrm{Recall}_i(M^{\mathrm{block}}) \le \frac{m}{2B}. \tag{19}$$

*Proof.* Each selected block $\mathcal{B}_r$ contributes $B$ tokens to the selected set $\widehat{I}_i$. Under budget $K$, the maximum number of blocks that can be selected is

$$S \le \frac{K}{B} = \frac{R_m}{B}.$$

In the worst-case alignment, each relevant segment $\mathcal{S}_c$ is split across two blocks, each containing at most $m/2$ important tokens. Thus, a single block contributes at most $m/2$ important tokens.

Hence, the total number of important tokens captured by any block-sparse pattern is upper bounded by

$$|I_i^\star \cap I_i^{\mathrm{block}}| \le S \cdot \frac{m}{2} \le \frac{R_m}{B} \cdot \frac{m}{2} = \frac{R_m^2}{2B}.$$

Dividing by $|I_i^\star| = R_m$ yields

$$\mathrm{Recall}_i(M^{\mathrm{block}}) = \frac{|I_i^\star \cap I_i^{\mathrm{block}}|}{|I_i^\star|} \le \frac{R_m^2}{2B} \cdot \frac{1}{R_m} = \frac{m}{2B},$$

which proves equation 19. $\qquad\square$

When blocks are much larger than the true semantic segments ($B \gg m$), the bound equation 19 implies that block-sparse recall under a fixed budget is at most $O(m/B)$, i.e., a small fraction of the oracle mass.

### B.6. Comparison and Discussion

Combining Propositions B.5 and B.6, we obtain,

$$\mathrm{Recall}_i(M^{\mathrm{DHSA}}) \ge 1 - \frac{2\delta}{m}, \qquad \mathrm{Recall}_i(M^{\mathrm{block}}) \le \frac{m}{2B}.$$

For realistic regimes where $m \ll B$ (semantic segments shorter than fixed blocks) and $\delta \ll m$ (accurate boundary prediction), we have

$$1 - \frac{2\delta}{m} \gg \frac{m}{2B},$$

i.e., DHSA achieves near-oracle recall, while any fixed block-sparse pattern under the same token budget can only capture an $O(m/B)$ fraction of the truly important keys in the worst case.

This analysis formalizes the intuitive advantage of *dynamic, semantics-aligned chunking*: by adapting chunk boundaries to semantic structure and using length-normalized chunk scores, DHSA selects units that closely match the true important regions in the attention map, whereas fixed block grids are oblivious to semantics and must include many uninformative tokens whenever segments are misaligned with block boundaries. Empirically, this prediction is consistent with our attention recall vs. oracle Top-$K$ results in Figure 12, where DHSA closely tracks the oracle while block-sparse baselines exhibit recall gaps under the same sparsity budget.

**Cost Analysis.**  We analyze the computational cost of **DHSA** as follows. Per layer, DHSA adds a linear $\mathcal{O}(L)$ pass for boundary prediction, forms chunk representations in $\mathcal{O}(L)$, computes chunk–chunk similarity in $\mathcal{O}(N_c^2)$ for $N_c$ chunks, and then performs token-level selection with a per-query budget $N_b$ in $\mathcal{O}(L\,N_b)$ without materializing a dense $L \times L$ matrix. We define the *token density* as $N_b/L$. In practice $N_b \ll L$ and $N_c \ll L$ (both are bounded by user budgets), so the dominant terms are $\mathcal{O}(L\,N_b)$, yielding near-linear scaling in $L$ for fixed $N_b$ and $N_c$. For comparison, block-sparse attention exhibits a complexity of $\mathcal{O}(L\,N_b) + \mathcal{O}(N_c^2)$, which is asymptotically similar to DHSA.

# C. Implementation Details

**Base Models.**  Our experiments use LLMs from three widely adopted open-source families: LLaMA, Qwen and Gemma. To study long-context inference under memory constraints, we select Llama-3.1-8B-Instruct (Dubey et al., 2024), Qwen2.5-3B-Instruct (Yang et al., 2024a), and gemma-2-2b-it (Team et al., 2024). The maximum context lengths are 128K for Llama-3.1-8B-Instruct, 32K for Qwen2.5-3B-Instruct, and 8K for gemma-2-2b-it. We apply *4-bit quantization* for Llama-3.1-8B-Instruct, and *torch.bfloat16* precision for Qwen2.5-3B-Instruct and Gemma-2-2B-it. In Llama-3.1-8B-Instruct and Qwen2.5-3B-Instruct, all layers employ global attention, whereas in gemma-2-2b-it, global attention is applied in every other layer with a sliding window of 4,096 tokens.

**Environment.**  All GPU-related experiments are conducted on a single NVIDIA RTX 3090 GPU (24 GB) running Ubuntu 22.04.4 LTS. The software environment includes Python 3.12, CUDA 12.4, PyTorch 2.5.1+cu124, and Transformers 4.52.3. For CPU experiments, we use an Intel Core 5 120U processor (10 cores, 12 threads, max frequency 1.40 GHz) running Windows 11. The software environment includes Python 3.11 and PyTorch 2.9.1+cpu, and Transformers 4.52.3.

**Backend Implementations.**  We provide two attention backends tailored to different deployment scenarios. Both backends take as input the *selected key indices* $\{\mathcal{I}_i\}_{i=0}^{N_r-1}$ produced by DHSA's routing step (Algorithm 1), and compute sparse causal attention for each query row block. The SDPA backend is broadly compatible with diverse model families (e.g., Gemma 2/3) and platforms, including CPU. The tiled online-softmax backend is more efficient on supported GPUs, as it follows a FlashAttention-style streaming softmax that avoids materializing attention matrices. Together, these backends cover a wide range of practical settings.

**1) PyTorch SDPA backend (Algorithm 2).** We implement this backend in PyTorch with Hugging Face Transformers, using PyTorch's scaled dot-product attention (SDPA) as the compute primitive. For each query row block $\mathbf{Q}_i = \mathbf{Q}[q_s : q_e]$, we gather the selected key/value states $\mathbf{K}_{sel} = \mathbf{K}[\mathcal{I}_i]$ and $\mathbf{V}_{sel} = \mathbf{V}[\mathcal{I}_i]$. We then construct an exact causal mask from absolute token positions and invoke $\mathrm{SDPA}(\mathbf{Q}_i, \mathbf{K}_{sel}, \mathbf{V}_{sel}, \mathbf{Mask})$ to obtain the output $\mathbf{O}[q_s : q_e]$. This backend supports fine-grained (token-level) selection and is portable across GPU and CPU environments. In practice, we compute attention in tiles and reuse buffers to bound memory when $|\mathcal{I}_i|$ is large.

**2) Tiled online-softmax backend (Algorithm 3).** For higher efficiency on supported GPUs, we also provide a memory-efficient tiled backend that streams over selected keys without materializing attention probabilities. For each row block, we iterate over the selected indices $\mathcal{I}_i$ in column tiles of size $B_c$. For each tile, we compute scores $\mathbf{S} = \mathbf{Q}_i \mathbf{K}_{tile}^\top / \sqrt{d}$, apply an exact causal mask using absolute positions, and update the normalization statistics $(m, \ell)$ and weighted sum $\mathbf{O}_{sum}$ via online softmax. After processing all tiles, we normalize $\mathbf{O}_{sum}$ by $\ell$ to produce $\mathbf{O}[q_s : q_e]$. This backend reduces peak memory and improves throughput by using tiled computation and streaming softmax accumulation.

In our implementation, we set the row block size and column tile size to $B_r = 128$ and $B_c = 128$, respectively, which aligns well with GPU warp-level execution and is consistent with common FlashAttention configurations.

**Additional Memory Optimizations.** For very long input sequences (e.g., $\geq$ 64K tokens), we avoid materializing large intermediate matrices by using tiled computation and promptly releasing temporary buffers. We also tile the boundary predictor MLP to reduce memory overhead on extremely long inputs. When applicable, we cache and reuse routing results (e.g., boundary indices and $\{\mathcal{I}_i\}$) to amortize overhead. Overall, the dominant attention computation scales with the selected keys, reducing the per-layer attention cost from $O(L^2)$ to approximately $O(L \cdot N_b)$ (up to constant factors), while preserving exact causal attention semantics.

**Baselines.** For a fair comparison, we use the same token density $N_b/L$ (6.25%, 12.5%, 25%) for all methods. For each baseline, we follow the hyperparameter choices recommended in the original papers or official implementations. To ensure stable results, all experiments use greedy decoding.

Specifically, the hyperparameters of each baseline are configured as follows:

1. **StreamingLLM** (Xiao et al., 2024), corresponding to the *A-shape* pattern. We allocate 20% of preserved keys as global tokens and 80% as local windows;

2. **StreamingLLM w/ dilated** (Beltagy et al., 2020), which applies dilated local windows with fixed intervals. We use 20% global tokens and 80% dilated windows with an interval of 1;

3. **StreamingLLM w/ strided** (Child, 2019), which combines local windows with dilated attention. We use 20% global tokens, 40% local windows, and 40% dilated windows with an interval of 1;

4. **MInference** (Jiang et al., 2024), which supports multiple sparsity patterns. We evaluate its Vertical-Slash configuration because the A-shape and block-sparse patterns are already covered by other baselines, and the original MInference ablation shows that Vertical-Slash alone performs close to the full configuration. We allocate 50% of preserved tokens to vertical-line tokens and 50% to slash-line tokens, with $last\_q = 64$.

5. **Block-Sparse Attention** (Guo et al., 2024), which selects tokens via blockwise similarity but lacks dynamic chunking and enhanced block representations compared to DHSA. We use a block size of 128 in all experiments.

6. **DuoAttention** (Xiao et al., 2025), which selects retrieval heads and streaming heads for a sparse KV cache. The head mask is learned during training, and we adjust the sparsity ratio in our experiments to match the desired density.

7. **SeerAttention** (Gao et al., 2024), which learns attention gates on top of block-sparse attention, with the key component being prediction of the block mask. Since it does not support a fixed token budget for sparse prefill, we directly set the non-zero ratio to match the target density.

8. **Quest** (Tang et al., 2024), which compresses the KV cache to lower memory footprint and improve decode-time efficiency. Given a token budget, it restricts each query to attend to that number of tokens during decoding. We set the token budget to correspond to the same effective density as in our setting.

For StreamingLLM and its variants, MInference, Block-Sparse Attention, and Quest, we configure the patterns so that each query selects approximately the same number of tokens implied by the token density (up to small discrepancies due to block- or window-based selection). For SeerAttention, the non-zero ratio is chosen to match the same global density, though the exact number of tokens per query can vary. DuoAttention effectively uses more tokens than other methods, since its streaming heads still attend to attention sinks and a fixed set of recent tokens. Note that for DuoAttention, SeerAttention, and Quest, we conduct experiments only on models they officially support (e.g., Llama-3.1-8B-Instruct) to ensure a fair comparison.

**Needle-In-A-Haystack.** We evaluated different baselines, starting with an assessment of the models' long-context processing capabilities through the needle-in-a-haystack test (Cai et al., 2025). This benchmark evaluates a model's ability to locate a target sentence (the needle) within a long context and is widely used for long-context language modeling. Our setup used context lengths ranging from 1K to 64K tokens and depth ranges from 0% to 100% (interval of 10%). The prompt format was: `<|im_start|> This is a very long story book: <book> {context} </book>. Based on the content of the book, Question: {retrieval_question} Answer:` with the needle sentence "The best thing to do in San Francisco is eat a sandwich and sit in Dolores Park on a sunny day." and the corresponding retrieval question "The best thing to do in San Francisco is:". We used ROUGE as the evaluation metric, and visualized results with green indicating correct and red indicating incorrect predictions.

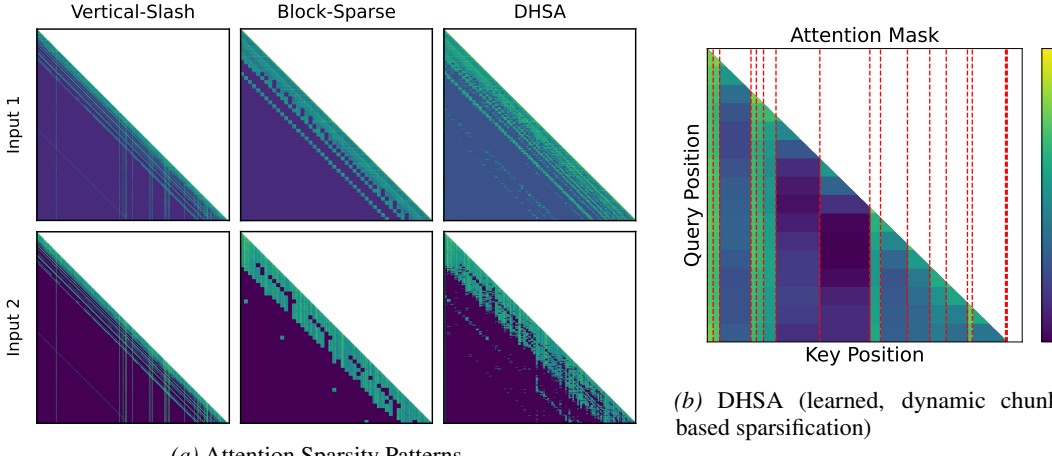

*(a)* Attention Sparsity Patterns

*(b)* DHSA (learned, dynamic chunk-based sparsification)

*Figure 11.* (a) Attention sparsity visualization for LLaMA-3.1-8B (4-bit). Existing methods such as Vertical-Slash and Block-Sparse enforces rigid regions. In contrast, DHSA produces adaptive masks across inputs. (b) DHSA introduces dynamic chunking (red dashed lines), offering greater flexibility and yielding better performance.

**LongBench.** LongBench (Bai et al., 2024) comprises 16 datasets designed to evaluate different aspects of long-context processing, including single-document QA, multi-document QA, summarization, few-shot learning, synthetic tasks, and code completion. Each task consists of multiple datasets: single-document QA includes NarrativeQA, Qasper, and MultiFieldQA_en; multi-document QA includes HotpotQA, 2WikiMultihopQA, and MuSiQue; summarization includes GovReport, QMSum, and MultiNews; few-shot learning includes TREC, TriviaQA, and SAMSum; synthetic tasks include PassageCount and PassageRetrieval_en; and code completion includes LCC and RepoBench-P. Average dataset lengths range from 1.2K to 18K tokens, and performance is measured using task-specific metrics (F1, ROUGE-L, accuracy, and edit similarity). We report the average score per task across its datasets. Since each model has a maximum context length $C_{max}$, for inputs exceeding this length we retain the first 1K tokens and the last $(C_{max} - 1)$ K tokens. To fit *Llama-3.1-8B-Instruct* (4-bit) on a single NVIDIA RTX 3090 GPU (24 GB), we set its $C_{max}$ to 48K.

**RULER.** RULER (Hsieh et al., 2024) is a controlled long-context benchmark that measures performance across different context lengths and task types. We evaluate on all RULER tasks and report the overall average from 4K to 48K tokens. To provide a more fine-grained analysis of distributed-evidence settings, we additionally report task-level breakdowns for `QA-1`, `QA-2`, and `VT`, which require retrieving or aggregating information from multiple context spans.

# D. Additional Results

**Overlap with Top-K Attention** To assess how well different sparse attention patterns preserve important information, we compute the overlap between each method's retained keys and the Top-K attention targets over the last 4K tokens of a 32K-context sequence. We extract attention scores from all heads and layers and report the average overlap to obtain a stable, model-wide estimate. Experiments are conducted using Llama-3.1-8B (4-bit) on the Needle-in-a-Haystack task. As shown in Figure 12, DHSA consistently achieves the highest overlap across all retention budgets, while other sparse patterns recover fewer important tokens. Overall, DHSA allocates its sparsity budget more effectively, providing the closest approximation to dense attention in long-context retrieval.

**Needle-In-A-Haystack.** As shown in Figure 1a, our method reliably retrieves information placed at different positions across context windows ranging from 1K to 100K tokens. We further evaluate several

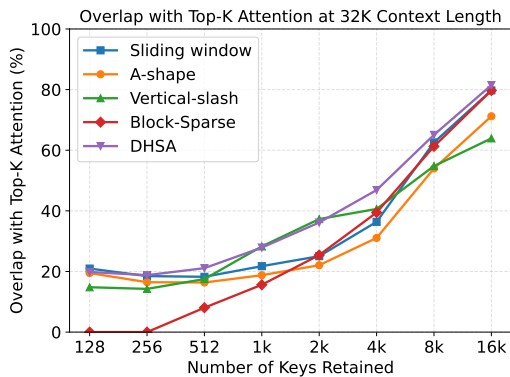

*Figure 12.* Overlap with Top-K attention as a function of the number of retained keys at 32K context length.

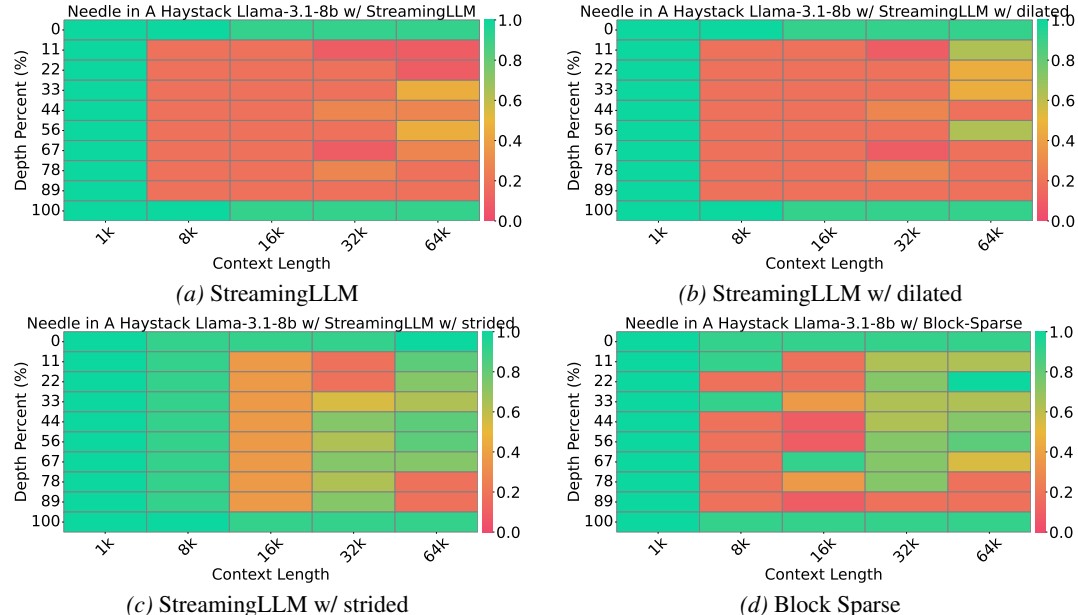

*(a) StreamingLLM*  *(b) StreamingLLM w/ dilated*

*(c) StreamingLLM w/ strided*  *(d) Block Sparse*

*Figure 13.* Needle In A Haystack results on Llama-3.1-8B (4-bit) with a token density of 6.25%.

*Table 6.* Task-level performance of different methods on selected RULER tasks (token density = 12.5%). The reported overall averages are computed over all RULER tasks.

| Method | 4K | | | | 8K | | | | 16K | | | | 32K | | | |
|---|---|---|---|---|---|---|---|---|---|---|---|---|---|---|---|---|
| | QA-1 | QA-2 | VT | Avg. | QA-1 | QA-2 | VT | Avg. | QA-1 | QA-2 | VT | Avg. | QA-1 | QA-2 | VT | Avg. |
| *Llama-3.1-8B-Instruct (4-bit)* | 83.8 | 72.0 | 100.0 | 95.8 | 78.5 | 72.0 | 99.6 | 93.8 | 72.2 | 66.0 | 100.0 | 93.2 | 70.5 | 66.0 | 67.6 | 81.5 |
| BlockSparse | 23.0 | 32.0 | 85.2 | 60.3 | 37.2 | 34.0 | 84.0 | 65.4 | 33.7 | 44.0 | 82.4 | 69.3 | 24.2 | 28.0 | 24.0 | 53.0 |
| **DHSA** | **74.3** | **65.8** | **99.4** | **92.1** | **69.1** | **63.3** | **97.6** | **88.5** | **63.5** | **61.1** | **97.9** | **88.7** | **59.2** | **56.8** | **61.4** | **76.2** |

baselines, including StreamingLLM (Xiao et al., 2024), StreamingLLM w/ dilated (Beltagy et al., 2020), StreamingLLM w/ strided (Beltagy et al., 2020), MInference (Jiang et al., 2024) and Block Sparse Attention (Guo et al., 2024). For all methods, the context length ranges from 1K to 64K tokens, and the token density (defined as the number of preserved keys divided by the context length) is fixed at 6.25%. For the 1K-token case, we preserve 512 keys, since 64 tokens (6.25% of 1024) are insufficient to maintain retrieval reliability.

From Figure 13, we observe that baselines such as StreamingLLM and Block Sparse Attention, while effective in reducing latency, suffer a sharp performance drop once the critical information lies outside their restricted attention ranges. This limitation highlights the trade-off between efficiency and accuracy in fixed or heuristic sparsity patterns, which fail to adapt to varying token distributions. In contrast, our method dynamically adjusts attention allocation, enabling it to maintain robust retrieval performance across diverse positions and context lengths while still achieving computational efficiency.

**Hyperparameter Sensitivity.** We analyze the sensitivity of DHSA to the boundary-detection window size $w$ and the NMS window size on Gemma-2-2B-it at a token density of 12.5%. All other settings are kept fixed. As shown in Table 7, performance is best at moderate values and degrades only when the window becomes too small or too large. This trend is consistent with the intuition that overly small windows provide insufficient local evidence, while overly large windows over-smooth boundary signals. Based on validation-set performance, we use the same boundary-detection and NMS hyperparameters across model families.

*Table 7.* Sensitivity analysis of boundary-detection and NMS window sizes with Gemma-2-2b-it (token density = 12.5%).

| Context window size ($w$) | 1 | 2 | 4 | 8 | 16 | 32 | | |
|---|---|---|---|---|---|---|---|---|
| Score | | 22.8 | 23.6 | **25.4** | 24.8 | 24.1 | 23.5 | |

| NMS window size | 0 | 1 | 2 | 4 | 8 | 16 | 32 | 64 | 128 |
|---|---|---|---|---|---|---|---|---|---|
| Score | 23.8 | 24.2 | 24.6 | 25.1 | **25.4** | 25.0 | 24.4 | 23.6 | 22.5 |

*Table 8.* Performance of different methods on LongBench with larger and higher-precision backbones (token density = 12.5%). All baselines are evaluated under matched sparsity settings.

| Methods | Single Doc QA | Multi Doc QA | Summ. | Few-shot Learning | Synth. | Code | Avg. |
|---|---|---|---|---|---|---|---|
| *Llama-3.1-8B-Instruct (BF16)* | 22.9 | 16.9 | 28.6 | 71.0 | 49.4 | 60.3 | 41.5 |
| BlockSparse | 15.3 | 8.8 | 21.6 | 64.9 | 40.4 | 49.5 | 33.4 |
| **DHSA** | **19.1** | **16.2** | **26.1** | **71.5** | **51.3** | **61.4** | **40.4** |
| *Qwen2.5-14B-Instruct (4-bit)* | 14.3 | 9.9 | 22.8 | 69.8 | 48.6 | 53.5 | 36.5 |
| BlockSparse | 9.9 | 6.4 | **21.7** | 62.0 | 11.3 | 30.6 | 23.6 |
| **DHSA** | **12.5** | **10.2** | **21.7** | **70.1** | **43.0** | **48.1** | **34.3** |

*Table 9.* TTFT comparison on LongBench with LLaMA-3.1-8B-Instruct (4-bit). We report representative datasets and the average over 16 datasets.

| Dataset | Avg. Tokens | Dense FA2 (s) | DHSA (s) |
|---|---|---|---|
| NarrativeQA | 29,869 | 10.73 | 5.37 |
| HotpotQA | 12,854 | 3.71 | 2.04 |
| Musique | 15,617 | 4.59 | 2.32 |
| QMSum | 13,917 | 4.05 | 2.15 |
| PassageCount | 14,970 | 4.38 | 2.26 |
| RepoBench-P | 10,818 | 3.07 | 1.84 |
| Avg. | – | 3.28 | 1.88 |

*Table 10.* Component-level breakdown for DHSA with LLaMA-3.1-8B-Instruct (4-bit). We report average time at different context lengths.

| Component | 8K (ms) | 16K (ms) | 32K (ms) |
|---|---|---|---|
| Boundary prediction | 45 | 97 | 201 |
| Routing / chunk selection | 237 | 379 | 944 |
| Sparse attention kernel | 128 | 316 | 939 |
| DHSA TTFT | 1,550 | 2,360 | 5,830 |
| Dense FA2 TTFT | 2,170 | 4,710 | 11,680 |

**Task-level RULER Results.** Table 6 reports task-level RULER (Hsieh et al., 2024) results for `QA-1`, `QA-2`, and `VT` under controlled context lengths. These tasks require aggregating information from distributed context spans and therefore provide a more fine-grained test beyond single-needle retrieval. DHSA consistently outperforms BlockSparse across all selected tasks and context lengths under the same token-density setting, with particularly large gains on `QA-1` and `QA-2`. The task-level results confirm that the aggregate gains reported in Table 5 are not driven by a single task, but reflect DHSA's ability to preserve multiple relevant regions under distributed evidence patterns.

**End-to-end Latency and Overhead Breakdown.** We report additional latency results in Tables 9 and 10. First, we measure TTFT on LongBench (Bai et al., 2024) with LLaMA-3.1-8B-Instruct (4-bit). DHSA reduces the average TTFT over 16 datasets from 3.28s with dense FA2 attention to 1.88s. Second, we provide a component-level breakdown of DHSA overhead, including boundary prediction, routing/chunk selection, and the sparse attention kernel. Although routing introduces additional overhead, the savings from sparse attention dominate at longer context lengths, leading to lower overall TTFT than dense FA2.

**Ablation Study.** To assess the contribution of different components in DHSA, we evaluate three variants: 1) DHSA without robust chunk representation; 2) DHSA without dynamic chunking; 3) DHSA without both dynamic chunking and robust chunk representation, which reduces the method to standard block-sparse attention. The results in Table 4 demonstrate that dynamic chunking plays a critical role in performance, as it adaptively partitions sequences based on semantic boundaries rather than relying on fixed templates. This flexibility allows the model to better capture long-range dependencies and context-specific structures, leading to consistent improvements across tasks. In addition, robust chunk representation yields further gains by normalizing and refining block-level embeddings, thereby reducing the risk that important tokens are diluted by surrounding less informative ones. Together, these components enable DHSA to strike a more effective balance between sparsity and expressiveness, outperforming static sparse baselines and narrowing the gap to dense attention while maintaining efficiency.

Recall that our robust chunk representation for chunk $\mathbf{C}_k$ of length $L_k = b_{k+1} - b_k$ is defined as

$$\mathbf{q}_{\mathbf{C}_k} = \frac{\sqrt{L_k}}{L_k} \sum_{i=b_k}^{b_{k+1}-1} \mathbf{q}_i, \qquad \mathbf{k}_{\mathbf{C}_k} = \frac{\sqrt{L_k}}{L_k} \sum_{i=b_k}^{b_{k+1}-1} \mathbf{k}_i. \tag{20}$$

Equivalently, $\mathbf{q}_{\mathbf{C}_k} = \sqrt{L_k}\, \bar{\mathbf{q}}_{\mathbf{C}_k}$, where $\bar{\mathbf{q}}_{\mathbf{C}_k} = \frac{1}{L_k} \sum_{i=b_k}^{b_{k+1}-1} \mathbf{q}_i$ is the average of token queries in the chunk. When dynamic

chunking is removed, the boundaries $\mathcal{B}$ are fixed to uniform blocks of size $B$, i.e., $b_k = kB$ and each chunk $\mathbf{C}_k$ spans a contiguous block $[kB, (k+1)B)$. In this case, the chunk-level similarity matrix $\mathbf{S}_c = \mathbf{Q}_c \mathbf{K}_c^\top$ is proportional (up to a constant factor $B$) to the blockwise similarity used in standard block-sparse attention with block size $B$. Since this factor is uniform across all block pairs, it does not affect which blocks are selected under a fixed sparsity budget, and the variant without dynamic chunking degenerates to a conventional block-sparse pattern.

**Larger and Higher-precision Backbones.** To evaluate whether DHSA generalizes beyond the main 4-bit LLaMA-3.1-8B-instruct setting, we additionally test Llama-3.1-8B-Instruct in BF16 precision and Qwen2.5-14B-Instruct in 4-bit precision. DHSA operates at the attention level and therefore does not require architecture-specific modifications. As shown in Table 8, DHSA remains close to dense attention in overall accuracy while improving over BlockSparse under the same token-density setting. The latency results (Table 11) further show that DHSA consistently reduces prefill latency compared with dense FlashAttention-2, with larger savings at longer context lengths and lower token densities.

*Table 11.* Kernel-level prefill latency with larger and higher-precision backbones. Dense FlashAttention-2 is used as the reference baseline.

| Method | Density | 8K (ms) | 32K (ms) | 128K (ms) |
|---|---|---|---|---|
| *Llama-3.1-8B-Instruct (BF16)* | – | 8.7 | 135.2 | 2297.3 |
| | 6.25% | 3.4 | 20.2 | 214.5 |
| DHSA | 12.5% | 4.0 | 29.3 | 360.0 |
| | 25.0% | 5.0 | 46.9 | 660.0 |
| *Qwen2.5-14B-Instruct (4-bit)* | – | 10.7 | 170.2 | 2873.2 |
| | 6.25% | 4.2 | 25.4 | 270.3 |
| DHSA | 12.5% | 5.0 | 37.0 | 454.1 |
| | 25.0% | 6.2 | 58.7 | 831.7 |

---

**Algorithm 1** DHSA Sparsity Prediction

---

**Require:** Query, key matrices $\mathbf{Q}, \mathbf{K} \in \mathbb{R}^{L \times d}$ of $L$ prompt tokens, prompt chunk boundaries $\{b_j\}_{j=0}^{N_c}$, row block size $B_r$, per-row-block selected-key budget $N_b$.

**Ensure:** Selected key-token index lists for row blocks $\{\mathcal{I}_i\}_{i=0}^{N_r-1}$.

1:  Partition queries into $N_r = \lceil L/B_r \rceil$ row blocks $\mathbf{Q}_i = \mathbf{Q}[iB_r : (i+1)B_r)$
2:  Partition keys into $N_c$ chunks $\mathbf{K}_j = \mathbf{K}[b_j : b_{j+1})$
3:  Compute row-block query representations $\mathbf{q}_i = \sqrt{B_r} \cdot \text{AvgPool}(\mathbf{Q}_i)$ for $i = 0, \ldots, N_r - 1$
4:  Compute chunk-level key representations $\mathbf{k}_j = \sqrt{b_{j+1} - b_j} \cdot \text{AvgPool}(\mathbf{K}_j)$ for $j = 0, \ldots, N_c - 1$
5:  Compute chunk-level similarity $\mathbf{S}_c \in \mathbb{R}^{N_r \times N_c}$ where $\mathbf{S}_c[i,j] = \langle \mathbf{q}_i, \mathbf{k}_j \rangle$
6:  **for** each row block $i = 0, \ldots, N_r - 1$ **in parallel do**
7:      $q_{\max} \leftarrow \min((i+1)B_r, \; L)$
8:      *# Rank key chunks by chunk-level similarity (descending order)*
9:      $\pi \leftarrow \text{ARGSORT}(\mathbf{S}_c[i,:], \text{descending=True})$
10:     *# Concatenate token indices following the ranked chunks*
11:     $\mathcal{I}_i \leftarrow [\,]$
12:     **for** each chunk id $j$ in order $\pi$ **do**
13:         $s \leftarrow b_j, \; e \leftarrow b_{j+1}$
14:         $s \leftarrow \min(s, q_{\max}), \;\; e \leftarrow \min(e, q_{\max})$
15:         **if** $s \geq e$ **then**
16:             **continue**
17:         **end if**
18:         Append indices $[s, s+1, \ldots, e-1]$ to $\mathcal{I}_i$
19:         **if** $|\mathcal{I}_i| \geq N_b$ **then**
20:             **break**
21:         **end if**
22:     **end for**
23:     *# Hard truncate to the token budget*
24:     $\mathcal{I}_i \leftarrow \mathcal{I}_i[: N_b]$
25:     $\mathcal{I}_i \leftarrow \text{SORT}(\mathcal{I}_i)$
26: **end for**
27: **return** $\{\mathcal{I}_i\}_{i=0}^{N_r-1}$

---

---

**Algorithm 2** DHSA Sparse Attention (SDPA Backend)

---

**Require:** Query, key, value matrices $\mathbf{Q}, \mathbf{K}, \mathbf{V} \in \mathbb{R}^{L \times d}$, row block size $B_r$, selected key-token indices $\{\mathcal{I}_i\}_{i=0}^{N_r-1}$.
**Ensure:** Output $\mathbf{O} \in \mathbb{R}^{L \times d}$.
1: $N_r \leftarrow \lceil L/B_r \rceil$
2: Initialize $\mathbf{O} \leftarrow \mathbf{0}$
3: **for** each row block $i = 0, \ldots, N_r - 1$ **in parallel do**
4:      $q_s \leftarrow iB_r, \;\; q_e \leftarrow \min((i+1)B_r, \; L)$
5:      $\mathbf{Q}_i \leftarrow \mathbf{Q}[q_s : q_e] \in \mathbb{R}^{(q_e - q_s) \times d}$
6:      $\mathcal{I} \leftarrow \mathcal{I}_i$
7:      $\mathbf{K}_{sel} \leftarrow \mathbf{K}[\mathcal{I}], \;\; \mathbf{V}_{sel} \leftarrow \mathbf{V}[\mathcal{I}]$
8:      *# Build causal mask from absolute positions*
9:      Initialize $\mathbf{Mask} \in \{0, -\infty\}^{(q_e - q_s) \times |\mathcal{I}|}$
10:      **for** each query row $u = 0, \ldots, q_e - q_s - 1$ **do**
11:        **for** each selected key index position $v = 0, \ldots, |\mathcal{I}| - 1$ **do**
12:          $\mathbf{Mask}[u, v] \leftarrow 0$ **if** $(q_s + u) \geq \mathcal{I}[v]$ **else** $-\infty$
13:        **end for**
14:      **end for**
15:      *# Compute attention via SDPA*
16:      $\mathbf{O}[q_s : q_e] \leftarrow \text{SDPA}(\mathbf{Q}_i, \mathbf{K}_{sel}, \mathbf{V}_{sel}, \mathbf{Mask})$
17: **end for**
18: **return** $\mathbf{O}$

---

---

**Algorithm 3** DHSA Sparse Attention (Tiled Online-Softmax Backend)

---

**Require:** Query, key, value $\mathbf{Q}, \mathbf{K}, \mathbf{V} \in \mathbb{R}^{L \times d}$, row block size $B_r$, selected key-token indices $\{\mathcal{I}_i\}_{i=0}^{N_r-1}$, column tile size $B_c$.
**Ensure:** Output $\mathbf{O} \in \mathbb{R}^{L \times d}$.
1: $N_r \leftarrow \lceil L/B_r \rceil$
2: Initialize $\mathbf{O} \leftarrow \mathbf{0}$
3: **for** each row block $i = 0, \ldots, N_r - 1$ **in parallel do**
4:      $q_s \leftarrow iB_r, \;\; q_e \leftarrow \min((i+1)B_r, \; L), \;\; B_r' \leftarrow q_e - q_s$
5:      $\mathbf{Q}_i \leftarrow \mathbf{Q}[q_s : q_e], \;\;\; \mathcal{I} \leftarrow \mathcal{I}_i$
6:      $\mathbf{m} \leftarrow -\infty \cdot \mathbf{1}_{B_r'}, \;\; \boldsymbol{\ell} \leftarrow \mathbf{0}_{B_r'}, \;\; \mathbf{O}_{sum} \leftarrow \mathbf{0}_{B_r' \times d}$
7:      **for** $t = 0$ to $|\mathcal{I}| - 1$ **step** $B_c$ **do**
8:        $\mathcal{J} \leftarrow \mathcal{I}[t : \min(t + B_c, |\mathcal{I}|)]$
9:        $\mathbf{K}_{tile} \leftarrow \mathbf{K}[\mathcal{J}], \;\; \mathbf{V}_{tile} \leftarrow \mathbf{V}[\mathcal{J}]$
10:        $\mathbf{S} \leftarrow \mathbf{Q}_i \mathbf{K}_{tile}^\top / \sqrt{d}$
11:        Apply causal mask: $\mathbf{S}[u, v] \leftarrow -\infty$ if $(q_s + u) < \mathcal{J}[v]$ for all $u, v$
12:        *# Online softmax update*
13:        $\mathbf{m}_{new} \leftarrow \max(\mathbf{m}, \text{rowmax}(\mathbf{S}))$
14:        $\boldsymbol{\alpha} \leftarrow \exp(\mathbf{m} - \mathbf{m}_{new})$
15:        $\mathbf{P} \leftarrow \exp(\mathbf{S} - \mathbf{m}_{new}[:, None])$
16:        $\boldsymbol{\ell} \leftarrow \boldsymbol{\ell} \odot \boldsymbol{\alpha} + \text{rowsum}(\mathbf{P})$
17:        $\mathbf{O}_{sum} \leftarrow \mathbf{O}_{sum} \odot \boldsymbol{\alpha}[:, None] + \mathbf{P}\mathbf{V}_{tile}$
18:        $\mathbf{m} \leftarrow \mathbf{m}_{new}$
19:      **end for**
20:      **for** $u = 0$ to $B_r' - 1$ **do**
21:        $\mathbf{O}[q_s + u, :] \leftarrow \mathbf{O}_{sum}[u, :] / \ell[u]$
22:      **end for**
23: **end for**
24: **return** $\mathbf{O}$

---

