# OpenReview forum: "Long-Context Modeling with Dynamic Hierarchical Sparse Attention for Memory-Constrained LLM Inference"
_ICML.cc/2026/Conference — ICML 2026 spotlight_

### Official Review · Reviewer_bT3j · 2026-03-03

**Soundness:** 4
**Presentation:** 3
**Significance:** 3
**Originality:** 3
**Overall Recommendation:** 5
**Confidence:** 4

**Summary:**

This paper proposes Dynamic Hierarchical Sparse Attention (DHSA), a content-adaptive sparse attention mechanism for long-context LLM inference under memory constraints. The method dynamically segments the input sequence into variable-length chunks using a learned boundary predictor operating on token keys. Chunk-level key representations are constructed via length-normalized pooling and chunk–chunk similarity scores are computed for query chunks with fixed size partition, and select top key chunks upto a certain budget. Experiments on Needle-in-a-Haystack and LongBench show improved accuracy–efficiency trade-offs compared to existing sparse attention baselines.

**Compliance With Llm Reviewing Policy:**

Affirmed.

**Final Justification:**

The core idea of dynamically partitioning tokens for sparse attention using a learned boundary predictor is intuitive and well-motivated. My main concern was that relying on a trained MLP limits generalizability across domains and architectures. The rebuttal provided additional RULER results and hyperparameter sensitivity analysis, but these do not directly address domain generalization. That said, the paper is technically solid, so my recommendation is accept.

**Key Questions For Authors:**

- What are the exact experimental settings (e.g., sparsity level, kernel, batch size, task) used for Figure 5?
- Can the authors report end-to-end speedup (not just kernel-level prefill speedup) for LongBench tasks?
- The experiments focus on LLaMA-3.1-8B with 4-bit quantization. How does DHSA scale to larger models or higher-precision inference?
- During evaluation, do all baselines use full dense decoding, or are sparse mechanisms also applied beyond prefill? Clarifying this would help assess the fairness of the reported results.
- Why are queries processed using fixed-size partitions? Could query boundaries instead be derived from key boundaries, or predicted using a separate boundary model, and what trade-offs would this introduce?

**Limitations:**

The paper does not explicitly discuss limitations or potential negative societal impacts, and there are no obvious negative societal implications beyond standard considerations related to the deployment of LLMs.

**Strengths And Weaknesses:**

**Strengths**
- The paper extends block-sparse attention by introducing a dynamic, content-aware mechanism for block boundary formation. This design choice is novel and well motivated.
- The proposed method shows consistent average accuracy improvements on LongBench.

**Weaknesses**
- DHSA relies on a learned boundary prediction module, trained offline with specific architectural and hyperparameter choices (e.g., window size, non-maximum suppression). While the predictor is lightweight, it introduces additional complexity, and its generalization across different model sizes and architectures remains unclear. A more systematic sensitivity analysis of these hyperparameters would help assess robustness and guide practical deployment.
- The paper does not provide a direct comparison against an oracle dynamic sparse attention baseline, such as Top-K token selection based on dense attention. It is therefore unclear how well DHSA approximates the optimal query–key interaction set (e.g., overlap with dense Top-K similarities under the same token budget).
- The comparison section is extensive but sometimes confusing. For example, MInference supports multiple sparsity patterns, yet only the vertical-slash pattern is evaluated, with this choice described only in the appendix. In this case, it would be clearer to compare methods explicitly by sparsity pattern rather than by method name, or to clarify how the selected configuration represents the full MInference approach.
- All reported speedups are kernel-level prefill speedups. The paper does not report end-to-end speedup on LongBench tasks, nor does it clearly quantify the overhead introduced by the boundary predictor and routing steps.

---

> ### Author Rebuttal · Authors · 2026-03-30
>
> We thank the reviewer for the valuable feedback. Below, we address the concerns and questions one by one.
> ## Response to Weakness 1
> Please refer to our response to Weakness 4 in Reviewer 435e for a detailed hyperparameter sensitivity analysis.
> ## Response to Weakness 2
> Our current paper already includes two analyses in this direction: Figure 2(b) reports attention recall across sparsity patterns, where DHSA consistently achieves higher recall and is closer to oracle Top-K selection; Figure 12 reports overlap with Top-K attention, where DHSA consistently achieves the highest overlap.
> ## Response to Weakness 3
> We report results for the vertical-slash pattern because A-shape and block-sparse patterns are already covered by other baselines. In addition, the ablation study in the original MInference paper (Table 4) shows that “w/ only vertical-slash” performs very close to full MInference. We will modify the method name accordingly in the revision.
> ## Response to Weakness 4
> We already report end-to-end TTFT in Figure 1(b). We will add end-to-end TTFT on LongBench and overhead breakdown with LLaMA-3.1-8B (4-bit) in the revision.
> | Dataset | Avg Tokens | Dense FA2 TTFT (s) | DHSA TTFT (s) |
> |---|---:|---:|---:|
> | narrativeqa | 29869 | 10.73 | 5.37 |
> | hotpotqa | 12854 | 3.71 | 2.04 |
> | musique | 15617 | 4.59 | 2.32 |
> | qmsum | 13917 | 4.05 | 2.15 |
> | passage_count | 14970 | 4.38 | 2.26 |
> | repobench-p | 10818 | 3.07 | 1.84 |
> | Average (over 16 datasets) | - | 3.28 | 1.88 |
>
> | Component | 8K Avg time (ms) | 16K Avg time (ms) | 32K Avg time (ms) |
> |---|---:|---:|---:|
> | Boundary prediction | 45 | 97 | 201 |
> | Routing / chunk selection | 237 | 379 | 944 |
> | Sparse attention kernel | 128 | 316 | 939 |
> | DHSA TTFT | 1550 | 2360 | 5830 |
> | Dense FA2 TTFT | 2170 | 4710 | 11680 |
> ## Response to Question 1
> Figure 5 is produced under the following settings:
> - Model: LLaMA-3.1-8B-Instruct (4-bit)
> - Accuracy metric: LongBench average score
> - Speed metric: LongBench average speedup
> - Token densities: 6.25%, 12.5%, and 25.0%
> - Platform: a single RTX 3090 (24 GB)
> - Batch size: 1
>
> Each point in Figure 5 corresponds to a specific method evaluated at one of these token densities.
> ## Response to Question 2
> Please refer to our response to Weakness 4.
> ## Response to Question 3
> DHSA operates purely at the attention level, and can be directly applied to higher-precision and larger backbones.
>
> For LLaMA-3.1-8B (BF16) with token density = 12.5%:
> | Method       | Single-Document QA | Multi-Document QA | Summarization | Few-shot Learning | Synthetic | Code | Avg. |
> |-------------|-------------------|------------------|---------------|------------------|----------|------|------|
> | Dense       | 22.9              | 16.9             | 28.6          | 71.0             | 49.4     | 60.3 | 41.5 |
> | BlockSparse | 15.3              | 8.8              | 21.6          | 64.9             | 40.4     | 49.5 | 33.4 |
> | DHSA        | 19.1              | 16.2             | 26.1          | 71.5             | 51.3     | 61.4 | 40.4 |
>
> | Method | Density | 8K latency (ms) | 32K latency (ms) | 128K latency (ms) |
> |---|---:|---:|---:|---:|
> | FA2 (Dense) | - | 8.7 | 135.2 | 2297.3 |
> | DHSA | 25% | 5.0 | 46.9 | 660.0 |
> | DHSA | 12.5% | 4.0 | 29.3 | 360.0 |
> | DHSA | 6.25% | 3.4 | 20.2 | 214.5 |
>
> For Qwen2.5-14B (4-bit) with token density = 12.5%:
> | Method       | Single-Document QA | Multi-Document QA | Summarization | Few-shot Learning | Synthetic | Code | Avg. |
> |-------------|-------------------|------------------|---------------|------------------|----------|------|------|
> | Dense       | 14.3              | 9.9              | 22.8          | 69.8             | 48.6     | 53.5 | 36.5 |
> | BlockSparse | 9.9               | 6.4              | 21.7          | 62.0             | 11.3     | 30.6 | 23.6 |
> | DHSA        | 12.5              | 10.2             | 21.7          | 70.1             | 43.0     | 48.1 | 34.3 |
>
> | Method | Density | 8K latency (ms) | 32K latency (ms) | 128K latency (ms) |
> |---|---:|---:|---:|---:|
> | FA2 (Dense) | - | 10.7 | 170.2 | 2873.2 |
> | DHSA | 25% | 6.2 | 58.7 | 831.7 |
> | DHSA | 12.5% | 5.0 | 37.0 | 454.1 |
> | DHSA | 6.25% | 4.2 | 25.4 | 270.3 |
> ## Response to Question 4
> Please refer to our response to Question 3 in Reviewer 58Zn for detailed evaluation settings.
> ## Response to Question 5
> Queries are processed using fixed-size partitions as a deliberate efficiency–accuracy trade-off. Keeping query partitions fixed provides a regular computation structure for chunk-level routing and sparse execution, which makes the method substantially simpler and more efficient. In principle, dynamic query chunking may provide additional flexibility, but it would also introduce extra complexity: query–key chunk pairing becomes less regular, kernel efficiency becomes harder to preserve, and the overhead of boundary prediction increases. Our current design preserves most of the accuracy gains while keeping routing and execution efficient.

---

> > ### Author Rebuttal · Reviewer_bT3j · 2026-04-01
> >
> > The authors have adequately addressed my concerns. I am raising my score from 4 to 5 accordingly.

---

> > > ### Author Response · Authors · 2026-04-08
> > >
> > > Thank you very much for the careful reading and for revisiting the score. We sincerely appreciate your thoughtful feedback and consideration.

---

### Official Review · Reviewer_435e · 2026-03-09

**Soundness:** 3
**Presentation:** 1
**Significance:** 3
**Originality:** 3
**Overall Recommendation:** 4
**Confidence:** 4

**Summary:**

The paper introduces Dynamic Hierarchical Sparse Attention (DHSA), a drop-in sparse attention module that accelerates the prefill stage of long-context LLM inference under strict memory constraints while keeping the LLM backbone frozen. DHSA predicts content-adaptive sparsity at inference by using a boundary predictor model to segment the prompt into variable-length chunks and then encoding those chunks with a length-normalized pooling scheme to compute chunk-level similarities. These similarity scores are used to select the relevant key chunks for a given token budget, after which a sparse attention backend computes exact causal attention only over the selected keys. DHSA significantly reduces prefill computational costs, delivering up to a 10x speedup over dense FlashAttention-2 at a 128K context length, without degrading performance on benchmarks like Needle-in-a-Haystack and LongBench.

**Compliance With Llm Reviewing Policy:**

Affirmed.

**Final Justification:**

I maintain my rating of 4 (weak accept). The core idea of dynamic, content-adaptive chunking for sparse prefill attention is novel, and the speedups without backbone finetuning are appealing. The rebuttal was responsive, the authors committed to moving critical pipeline details into the main text, clarified the lightweight training cost of the boundary predictor, provided a helpful hyperparameter sensitivity analysis, and added RULER evaluations that extend evidence beyond retrieval-only tasks. That said, some concerns remain partially unresolved. The sequential batching requirement is a meaningful practical limitation, LongBench gains over competitive sparse methods look marginal once batching overhead is considered. The presentation of the paper is also a drawback, while the promised revisions should help, the original submission was very difficult to evaluate from the main text alone. Overall, the paper makes a solid contribution to an important problem.

**Key Questions For Authors:**

Please refer to the weaknesses section.

**Limitations:**

Yes

**Strengths And Weaknesses:**

### Strengths
1. The dynamic chunking of tokens based on the content is a novel idea. It allows the method to adapt to the attention patterns based on the input task.
2. The method does not require any finetuning of the original LLM backbone.
3. DHSA achieves significant prefill speedups over dense FlashAttention while maintaining similar accuracy.
4. The attention recall analysis shows why their data-driven block sparse attention is better than static sparse attention methods.

### Weaknesses

1. A lot of the critical details about the method, such as the boundary predictor architecture, the automatic labeling strategy, and the NMS inference, are pushed to the appendix. So by just reading the main text, it is very difficult to understand how this method works.

1. It's not clear whether the boundary predictor needs to be trained per LLM backbone since its labels come from the attention scores. Also the training details of the boundary predictor like compute and steps needed are missing.

1. The boundary predictor looks at the future tokens, so it might not be compatible with streaming or prefix processing.

1. Missing ablations on the hyperparameters like the window size for boundary detection and NMS window size. So it is unclear how to pick the best hyperparameters if using this method with different models.

1. Lacking benchmarks like RULER where the context length can be controlled to see if DHSA's accuracy starts to drift as the context length increases. The evaluations are done on NIAH, but that is a very simple task that's retrieval only.

1. DHSA requires sequential processing for batched sequences. While this avoids OOM, relying on sequential processing inherently limits the maximum hardware throughput you would normally get from parallelized batched operations.

---

> ### Author Rebuttal · Authors · 2026-03-30
>
> We thank the reviewer for the valuable feedback. Below, we address the concerns and questions one by one.
>
> ## Response to Weakness 1
>
> We thank the reviewer for pointing this out. We agree that several details that are important for understanding the full DHSA pipeline are currently described mainly in the appendix. In the revision, we will move a concise summary of the boundary predictor architecture, the automatic label construction procedure, and the boundary finalization step into the main text, while keeping the full implementation details in the appendix. We agree that summarizing these steps in the main paper would make the method substantially easier to follow and would improve reproducibility.
>
> ## Response to Weakness 2
>
> We thank the reviewer for raising this point. In our current implementation, the boundary predictor is trained per backbone family, rather than shared universally across different LLM backbones. This is intentional: the predictor operates directly on backbone-specific key representations, whose feature spaces differ across models. We will clarify this explicitly in the revision. Importantly, this does not require finetuning the LLM itself; only a lightweight standalone predictor is trained, while the backbone remains frozen. We view this as a lightweight model-specific calibration step, analogous to attaching a small auxiliary module to a frozen backbone.
>
> The training cost is also lightweight. We train the predictor on a single RTX 3090 for 10,000 steps. See Line 692 for data preparation and Line 706 for training acceleration details. For Gemma2-2b-it, the wall-clock time is 3.20 s/it, corresponding to a total training time of 8.89 hours.
>
> ## Response to Weakness 3
>
> We thank the reviewer for pointing this out. In the current design, the boundary predictor is used during the prefill stage, where the full prompt is already available. Therefore, using both left and right context for boundary prediction is consistent with our target setting. Our goal in this paper is prefill acceleration for long-input inference. We will clarify this scope more explicitly in the revision. Supporting streaming or prefix-only processing is an interesting extension, but is outside the scope of the current work.
>
> ## Response to Weakness 4
>
> We thank the reviewer for raising this point. During development, we select these hyperparameters on the validation set (see Line 692 for data preparation). Below we show the sensitivity analysis on Gemma-2-2b-it at a token density of 12.5%. All other settings are kept at their default values, and only the hyperparameter under study is varied.
>
> | window size for boundary detection (w) | 1 | 2 | 4 | 8 | 16 | 32 |
> |---|---:|---:|---:|---:|---:|---:|
> | Score | 22.8 | 23.6 | 25.4 | 24.8 | 24.1 | 23.5 |
>
> | NMS window size | 0 | 1 | 2 | 4 | 8 | 16 | 32 | 64 | 128 |
> |---|---:|---:|---:|---:|---:|---:|---:|---:|---:|
> | Score | 23.8 | 24.2 | 24.6 | 25.1 | 25.4 | 25.0 | 24.4 | 23.6 | 22.5 |
>
> These results show a clear and intuitive trend: performance is best at moderate values. In our experiments, we observe the same overall trend across different model families. As a result, we use the same boundary-detection and NMS hyperparameters across all models, which makes DHSA easy to deploy in practice.
>
> In the revision, we will add an explicit sensitivity analysis and clarify the hyperparameter selection procedure.
>
> ## Response to Weakness 5
>
> We thank the reviewer for this suggestion. In the paper, we already report results on NIAH and LongBench. While NIAH mainly evaluates retrieval-style localization, LongBench covers a broader set of long-context tasks beyond simple retrieval. To address the reviewer’s concern more directly, we additionally evaluate DHSA on RULER, where the context length can be explicitly controlled.
>
> Please refer to our response to Weakness 1 in Reviewer 58Zn for the results on RULER.
>
> As shown in the Table, DHSA consistently outperforms BlockSparse across context lengths on RULER. These results indicate that DHSA delivers stable accuracy gains as the context length increases, and provide stronger evidence beyond retrieval-only benchmarks.
>
> ## Response to Weakness 6
>
> We agree that sequential processing is not the maximum-throughput design. In the current work, we deliberately prioritize input-adaptive sparsity and memory robustness over enforcing a shared batch mask. Supporting efficient batched execution for fully dynamic sparse attention is challenging because each sequence induces different routed indices, making it difficult to apply a single shared sparse pattern across the batch.
>
> A promising future direction is to group sequences with similar routed patterns and partially share masks (or their union) within each group. We leave this as future work, since our goal here is to demonstrate practical prefill gains in memory-constrained settings, rather than optimize datacenter-scale maximum batch throughput.

---

> > ### Author Rebuttal · Reviewer_435e · 2026-03-31
> >
> > * The sequential processing for batched sequences reduces the practicality of this method. On LongBench the gains over other competitive sparse methods look marginal, so those baselines may win in practice once batching is accounted for.
> > * The added RULER scores are shown only up to 32K and mainly compared against BlockSparse (which is one of the weaker baselines), so it’s still unclear if DHSA has significant improvements over stronger baselines at controlled lengths.

---

> > > ### Author Response · Authors · 2026-04-03
> > >
> > > We thank the reviewer for the follow-up questions. Below, we address them one by one.
> > >
> > > ## Response to Question 1
> > >
> > > We respectfully note that practical batching is not a solved capability for existing sparse-attention baselines, especially when the effective sparse pattern differs across sequences. For example, in the original papers of MInference and SeerAttention, we are not aware of clear evidence demonstrating that they support heterogeneous batched sparse prefill.
> > >
> > > More broadly, once each sequence induces different sparse indices, batching becomes a general systems challenge rather than a DHSA-specific limitation. In the current work, DHSA focuses on memory-constrained hardware scenarios, where input-adaptive sparsity and memory robustness are prioritized. We view higher-throughput grouped batching as an important future direction (e.g., group sequences with similar routed patterns and partially share masks (or their union) within each group).
> > >
> > > ## Response to Question 2
> > >
> > > Our method focuses on memory-constrained hardware scenarios, and all experiments are conducted on a single RTX 3090 (24GB). To address this point more directly, we extend the controlled-length evaluation to 48K, the longest context length at which dense attention remains feasible in our setting, and we additionally include stronger sparse baselines beyond BlockSparse.
> > >
> > > As shown in the table below, DHSA achieves the best overall average among sparse baselines at longer contexts (e.g., 32K and 48K), while remaining competitive at shorter context lengths. These results suggest that DHSA retains a stronger accuracy–efficiency trade-off than prior sparse baselines.
> > >
> > > We also note that DuoAttention and Quest are primarily designed for decoding-stage acceleration and do not target the prefill bottleneck addressed in this work. In contrast, while MInference is a prefill-oriented method, our results show that it does not outperform FlashAttention-2 in kernel-level prefill speedup (see Figure 5 in our paper).
> > >
> > > **Table 1: Performance of LLaMA-3.1-8B (4-bit) on RULER (token density = 12.5%)**
> > > | Method | Overall Avg @ 4K | Overall Avg @ 8K | Overall Avg @ 16K | Overall Avg @ 32K | Overall Avg @ 48K |
> > > |---|---:|---:|---:|---:|---:|
> > > | Dense | 95.78 | 93.81 | 93.20 | 81.52 | 75.22 |
> > > | MInference | 91.68 | 88.38 | 88.64 | 74.68 | 64.15 |
> > > | BlockSparse | 60.34 | 65.38 | 69.34 | 52.98 | 43.07 |
> > > | DuoAttention | 92.03 | **89.26** | **89.65** | 72.92 | 69.13 |
> > > | SeerAttention | 86.88 | 89.03 | 89.07 | 74.97 | 53.56 |
> > > | Quest | 87.65 | 86.02 | 87.18 | 73.85 | 68.20 |
> > > | DHSA | **92.13** | 88.46 | 88.73 | **76.17** | **71.51** |

---

### Official Review · Reviewer_58Zn · 2026-03-12

**Soundness:** 3
**Presentation:** 4
**Significance:** 3
**Originality:** 3
**Overall Recommendation:** 4
**Confidence:** 4

**Summary:**

The paper proposes Dynamic Hierarchical Sparse Attention (DHSA), an input-adaptive framework designed to accelerate the prefill stage of long-context Large Language Models under strict memory constraints. To address the quadratic computational cost of dense attention, DHSA introduces a drop-in module that predicts token level attention sparsity dynamically without requiring fine tuning of the LLM backbone. The core innovation lies in a hierarchical routing mechanism: it first uses a lightweight boundary predictor to segment the input sequence into variable length semantic chunks, computes chunk level similarities, and then routes queries to a compact set of highly relevant key tokens. Empirical evaluations demonstrate that DHSA effectively preserves near-dense accuracy on benchmarks like Needle-in-a-Haystack and LongBench while offering significant latency speedups and memory reductions over static and block-sparse baselines.

**Compliance With Llm Reviewing Policy:**

Affirmed.

**Key Questions For Authors:**

1.On DHSA generalization: Since DHSA routing depends on semantic block assumptions, would Attention Recall collapse when handling highly dispersed code branches or multi-step math reasoning involving very short fragments?

2.Boundary Prediction Stability for OOD Data: The Encoder's training labels are derived from specific datasets. When encountering OOD data with significantly different structure which may cause dynamic chunking degenerate into random chunking?

3.The authors must explicitly clarify whether the autoregressive decoding phase inherits the DHSA sparsity mask established during the prefill stage. Many baseline methods perform decoding while operating on a strictly compressed KV cache. If DHSA abandons its sparse mask and defaults to full dense attention during decoding, comparing its accuracy against these heavily constrained baselines is fundamentally unfair.

**Limitations:**

yes

**Strengths And Weaknesses:**

Soundness:

- Strengths: The technical methodology is well-founded. The empirical evaluation is thorough, benchmarking against a strong suite of recent sparse attention methods (StreamingLLM, MInference, Block-Sparse, etc.) across multiple open-weight models (LLaMA-3.1-8B, Qwen2.5-3B, Gemma-2-2B).

- Weakness: The methodology relies heavily on the assumption that critical attention mass clusters into contiguous semantic segments. While this holds for retrieval and standard document QA, the paper lacks validation on tasks where attention patterns might be highly scattered or entangled (e.g., complex multi-step math or logic reasoning). Furthermore, the boundary predictor is trained using labels derived from specific datasets (TriviaQA, ChatQA2, etc.), which raises minor concerns about domain bias generalizing to out-of-distribution texts like code or structured data.

Presentation:

- Strengths: The paper is highly readable and well-structured. The narrative flows logically from the limitations of static sparsity to the design of the hierarchical routing mechanism.

- Weakness: Some crucial implementation details that affect reproducibility and real world performanc, such as the NMS window size and thresholding heuristic are relegated entirely to the appendix. Briefly mentioning these in the main text would improve clarity regarding how boundaries are finalized.

Significance:

- Strengths: The problem addressed is highly relevant. Serving long-context LLMs on constrained hardware is a major bottleneck for practitioners. By focusing on the prefill stage and offering an adaptable, plug-and-play solution that supports both CPU and GPU backends, this work provides immediate practical utility to the community.

- Weakness: Although DHSA can address the issues of existing full attention models, current mainstream models have shifted towards linear attention and sliding window attention. Additionally, some sparse attention mechanisms such as DSA are also widely used. Ultimately, these training-based methods will replace training-free approaches.

Originality:

While sparse attention and dynamic chunking are not entirely new concepts, DHSA combines them in a novel and effective way. Framing chunking as a 1D boundary detection problem and solving it via a lightweight, shared encoder with Non-Maximum Suppression is a creative adaptation of computer vision techniques to sequence modeling. This input adaptive approach represents a meaningful step forward from rigid, heuristic-based sparsity templates.

---

> ### Author Rebuttal · Authors · 2026-03-30
>
> We thank the reviewer for the valuable feedback. Below, we address the concerns and questions one by one.
>
> ## Response to Weakness 1 (Soundness)
>
> We agree that some long-context tasks require integrating evidence distributed across multiple distant spans, rather than relying on a single contiguous segment. DHSA does not assume that all useful evidence collapses into one contiguous region; instead, it assumes that sparse attention can often be well approximated by a small set of salient segments, which may be distributed across the sequence.
>
> To evaluate this more directly, we add experiments on RULER. Since DHSA targets prefill acceleration for long-input settings, we focus on reasoning-oriented long-context tasks rather than traditional short-input, long-output math/logic benchmarks. In particular, we report results on QA-1, QA-2, and variable tracking (VT), which require combining information from multiple distributed spans.
>
> As shown below, DHSA consistently outperforms BlockSparse on these reasoning-oriented subsets across context lengths. These results suggest that DHSA does not depend on a single contiguous segment, and can preserve multiple relevant regions even when evidence is more fragmented or distributed.
>
> **Table 1: Performance of LLaMA-3.1-8B (4-bit) on RULER (Token density = 12.5%)**
> | Context | Dense | BlockSparse | DHSA | DHSA vs BlockSparse |
> |---|---:|---:|---:|---:|
> | **Overall Avg @ 4K** | 95.78 | 60.34 | 92.13 | **+31.79** |
> | **Overall Avg @ 8K** | 93.81 | 65.38 | 88.46 | **+23.08** |
> | **Overall Avg @ 16K** | 93.20 | 69.34 | 88.73 | **+19.39** |
> | **Overall Avg @ 32K** | 81.52 | 52.98 | 76.17 | **+23.19** |
> |  |  |  |  |  |
> | **QA-1 @ 4K** | 83.83 | 23.00 | 74.34 | **+51.34** |
> | **QA-1 @ 8K** | 78.50 | 37.17 | 69.05 | **+31.88** |
> | **QA-1 @ 16K** | 72.17 | 33.67 | 63.54 | **+29.87** |
> | **QA-1 @ 32K** | 70.50 | 24.17 | 59.24 | **+35.07** |
> |  |  |  |  |  |
> | **QA-2 @ 4K** | 72.00 | 32.00 | 65.76 | **+33.76** |
> | **QA-2 @ 8K** | 72.00 | 34.00 | 63.31 | **+29.31** |
> | **QA-2 @ 16K** | 66.00 | 44.00 | 61.07 | **+17.07** |
> | **QA-2 @ 32K** | 66.00 | 28.00 | 56.77 | **+28.77** |
> |  |  |  |  |  |
> | **VT @ 4K** | 100.00 | 85.20 | 99.36 | **+14.16** |
> | **VT @ 8K** | 99.60 | 84.00 | 97.64 | **+13.64** |
> | **VT @ 16K** | 100.00 | 82.40 | 97.88 | **+15.48** |
> | **VT @ 32K** | 67.60 | 24.00 | 61.41 | **+37.41** |
>
> As for the generalizability of the boundary predictor, while it is trained on datasets such as TriviaQA and ChatQA2, it is designed to capture general structural patterns in attention (i.e., semantic boundaries) rather than task-specific semantics. As a result, it is expected to generalize beyond the training domains.
>
> Empirically, we evaluate DHSA across a diverse set of benchmarks with different data distributions, including Needle-in-a-Haystack, LongBench, and RULER. In particular, LongBench includes coding tasks. Despite this distribution shift, DHSA maintains strong performance and consistently outperforms static sparse baselines, suggesting that the learned boundary predictor does not overfit to the training domains.
>
> ## Response to Weakness 2 (Presentation)
>
> We thank the reviewer for pointing this out. In the revision, we will move a concise description of the boundary finalization pipeline into the main text.
>
> ## Response to Weakness 3 (Significance)
>
> We agree that architecture-level efficient attention and training-based sparse attention are important directions. However, DHSA addresses a complementary and practically important setting: accelerating already-trained long-context LLMs without modifying model weights. This is particularly valuable for open checkpoints and deployed systems, where retraining is often costly or infeasible. In this sense, training-free methods remain highly relevant even as model architectures continue to evolve, since they provide an immediate path to improving efficiency for existing models rather than requiring a new pretraining or finetuning pipeline.
>
> ## Response to Question 1
>
> Please refer to our response to Weakness 1.
>
> ## Response to Question 2
>
> Please refer to our response to Weakness 1.
>
> ## Response to Question 3
>
> For prefill-oriented methods (MInference, Block Sparse, SeerAttention, and DHSA), we use the standard evaluation setting adopted in prior work (MInference, SeerAttention): sparse prefill with standard dense decoding. For StreamingLLM, it was originally proposed for decoding-stage efficiency. We follow MInference and repurpose its sparse pattern for prefill-stage evaluation, again using sparse prefill with standard dense decoding. For other decoding-oriented methods (DuoAttention and Quest), we use their original implementations in their native setting, i.e., dense prefill with sparse decoding, and report them as reference points for final downstream accuracy. Since our work focuses on prefill acceleration, all reported latency results are prefill latency and speedup.

---

> > ### Author Rebuttal · Reviewer_58Zn · 2026-04-05
> >
> > Thank you for the thorough response addressing all my questions. I will maintain my current rate.

---

> > > ### Author Response · Authors · 2026-04-08
> > >
> > > Thank you again for the careful reading and thoughtful feedback. We appreciate your time and consideration.

---

### Official Review · Reviewer_d9x3 · 2026-03-12

**Soundness:** 4
**Presentation:** 4
**Significance:** 3
**Originality:** 3
**Overall Recommendation:** 6
**Confidence:** 4

**Summary:**

this paper propose dhsa, that first identify the chunk boundary with a lightweight mlp, and do pooling chunk-wise, then estimate the similarity matrix at chunk-level, then we kinda masked out less important chunks in self-attention computation.

**Compliance With Llm Reviewing Policy:**

Affirmed.

**Key Questions For Authors:**

no

**Strengths And Weaknesses:**

strength

1. clever idea and impressive results. all previous methods use a fixed size chunk, even distillation methods such as seerattention. but this paper introduce an innovative idea to identify chunk boundary on the fly. the training of that boundary identification layer is lightweight and easy to train.

2. impressive results. although the proposed method did chunking online, but it outperform offline profiled method like MInference and fixed chunking method seerattention. achieving impressive sparse ratio and speedup.

3. solid experiments. comprehensive baselines and test set. extensive analysis.

4. nice presentation and easy to follow.

---

> ### Author Rebuttal · Authors · 2026-03-30
>
> Thank you for the encouraging feedback.
>
> Accelerating long-context modeling is an important and practical problem, especially in memory-constrained settings. We observed that existing static sparse methods cannot adapt well to task- or input-dependent variations, while recent dynamic approaches still rely on predefined templates or heuristics.
>
> To address this, we propose DHSA, a data-driven framework that predicts attention sparsity online while keeping the LLM backbone frozen. DHSA performs hierarchical routing by estimating importance at the chunk level and propagating it to token-level interactions, allowing it to preserve causally important dependencies while enabling efficient sparsification. Extensive experimental results show that DHSA provides an effective and adaptable solution for memory-constrained long-context LLM inference.
>
> We are glad that the reviewer appreciated both the idea and the implementation. Thank you again for the thoughtful and supportive review.

---

### Decision · Program_Chairs · 2026-04-30

**Decision:**

Accept (spotlight)

**Comment:**

This paper proposes DHSA (Dynamic Hierarchical Sparse Attention), a framework for reducing prefill latency in long-context LLM inference. The core idea is to dynamically partition the input sequence into variable-length semantic chunks using a lightweight boundary predictor, perform chunk-level similarity routing based on length-normalized chunk representations, and then propagate chunk-level importance scores to the token level under a fixed token budget. In this way, the method enables query-aware sparse attention while keeping the backbone LLM frozen.
The paper evaluates DHSA on Needle-in-a-Haystack, LongBench, and RULER, across several models including LLaMA-3.1-8B, Qwen2.5-3B, Gemma-2-2B, and Qwen2.5-14B. The results indicate a strong accuracy-efficiency trade-off. In particular, at 128K context length, the method achieves substantial prefill speedups over dense FlashAttention-2, while also enabling longer-context inference in memory-constrained environments.
Following the rebuttal, all reviewers expressed an overall positive stance. Several reviewers emphasized that the paper tackles an important and practical problem, and that the proposed combination of dynamic content-aware chunking and hierarchical routing is a meaningful contribution. Compared with static block-sparse methods, this design appears better aligned with the task- and input-dependent attention patterns that arise in long-context settings. Reviewers also viewed the empirical evaluation as solid (d9x3, 58Zn, 435e, bT3j), and the main concerns raised in the initial reviews were largely addressed during rebuttal.

Overall, I believe the paper makes a valuable contribution on an important problem, and I recommend acceptance.